# Short communication: Synchrotron-based elemental mapping of single grains to investigate variable infrared-radiofluorescence emissions for luminescence dating

Mariana Sontag-González[1,2*], Raju Kumar[3*], Jean-Luc Schwenninger[3], Juergen Thieme[1,4], Sebastian Kreutzer[5], Marine Frouin[1]

[1] Department of Geosciences, Stony Brook University, 255 Earth and Space Sciences Building, Stony Brook, NY 11794-2100, USA
[2] Department of Geography, Justus Liebig University Giessen, 35390 Giessen, Germany
[3] Research Laboratory for Archaeology and the History of Art, University of Oxford, Dyson Perrins Building, South Parks Road, OX1 3QY, Oxford, UK
[4] Institute for X-Ray Physics, Georg-August-University of Goettingen, Germany
[5] Institute of Geography, Ruprecht-Karl University of Heidelberg, 69120 Heidelberg, Germany

*These authors contributed equally to this work.

*Correspondence to*: Mariana Sontag-González (mariana.sontag-gonzalez@geogr.uni-giessen.de) or Raju Kumar (raju.kumar@arch.ox.ac.uk)

**Abstract.** During ionizing irradiation, potassium (K)-rich feldspar grains emit infrared (IR) light, which is used for infrared-radiofluorescence (IR-RF) dating. The late-saturating IR-RF emission centred at ~880 nm represents a promising tool to date Quaternary sediments. In the present work, we report the presence of individual grains in the K-feldspar density fraction displaying an aberrant IR-RF signal shape, whose combined intensity contaminates the sum signal of an aliquot composed of dozens of grains. Our experiments were carried out at the National Synchrotron Light Source (NSLS-II) at the submicron resolution X-ray spectroscopy (SRX) beamline. We analysed coarse (> 90 µm) K-feldspar bearing grains of five samples of different ages and origin in order to characterize the composition of grains yielding the desired or contaminated IR-RF emission. Using micro-X-ray-fluorescence (µ-XRF), we successfully acquired element distribution maps of up to 15 elements (<1 µm resolution) of sections of full grains previously used for IR-RF dating. In keeping with current theories of IR-RF signal production, we observed a trend between the relative proportions of Pb and Fe and the shape of the IR-RF signal: most grains with the desired IR-RF signal shape had high Pb and low Fe contents. Interestingly, these grains were also defined by high Ba and low Ca contents. Our study also represents a proof-of-concept for mapping the oxidation states of Fe using micro-X-ray absorption near-edge structure spectroscopy (µ-XANES) on individual grains. The high spatial resolution enabled by synchrotron X-ray spectroscopy makes it a powerful tool for future experiments to elucidate long-standing issues concerning the nature and type of defect(s) associated with the main dosimetric trap in feldspar.

## 1 Introduction

Geochronologic data provide essential information for understanding the rates of Earth's surface processes, environmental changes, and the evolution of life. Advances in dating techniques have fundamentally changed our capacity to piece together our evolutionary past over millions of years, with luminescence dating proving to be a powerful tool in this field as it applies to various types of sediments and contexts. The technique determines an age estimate for when mineral grains were last exposed to daylight or heat. Luminescence dating methods rely on the property of certain minerals to record the amount of radiation to which they have been exposed during burial and release energy when exposed to sunlight or high temperature (e.g., Aitken, 1985, 1998; Bateman, 2019). In the laboratory, the total amount of energy per unit mass stored in the mineral is measured (dose, with the unit Gy). The energy absorption rate per unit mass (dose rate, with the unit Gy $a^{-1}$) is derived from knowledge of the natural radioactivity surrounding the sampled sediments. The quotient of these two values (dose/dose rate) gives the burial age.

Of the two minerals routinely used for luminescence dating of sediments, quartz and potassium (K)-rich feldspar, the latter allows for the routine dating of older deposits of up to ~300 000 years or ~900 000 years (considering a dose rate of 3 Gy $ka^{-1}$ or 1 Gy $ka^{-1}$, respectively) using infrared stimulated luminescence (IRSL, Hütt et al., 1988). The datable upper age limit is given by the IRSL signal saturation after exposure to radiation doses around 900 Gy (see summary in Sec. 8.1 in Murari et al., 2021a). Over the past decades, different methods have been proposed to extend this upper age limit with varying degrees of success. The infrared-radiofluorescence (IR-RF) signal of K-feldspar is a promising candidate for such an extension. The RF signal arises from prompt radiative recombination of charge within crystalline materials during continuous exposure to ionizing radiation. The IR-RF emission at 880 nm (e.g., Kumar et al., 2018; Riedesel et al., 2021; Sontag-González et al., 2022) decreases in intensity with dose accumulation as the electron traps fill until saturation (Trautmann et al., 1999a). This saturation level constrains the time range over which IR-RF dating is applicable.

Murari et al. (2018) demonstrated that an accurate dose recovery of a known dose of 3600 Gy is possible (a dose recovery test is a laboratory performance check of the measurement protocol, and successful dose recovery is a prerequisite for any protocol). If we assume typical environmental dose rates of between 3 Gy $ka^{-1}$ and 1 Gy $ka^{-1}$, then IR-RF dating could produce age estimates ranging from 1.2 Ma to 3.6 Ma, which is around four times greater than the upper dating limit of conventional luminescence dating methods. However, more recent studies (Murari et al., 2021b; Kreutzer et al., 2022) indicated a dose saturation at around 1500 Gy, reducing the previously predicted temporal limit of IR-RF dating. Hence, the uncertainty surrounding its upper age limit remains and further studies on known-age samples are required to assess whether the sample/grain geochemistry influences the age limit. There is undoubtedly a gap in our current understanding of the IR-RF production processes in K-feldspar, and a revised conceptual model might be needed.

The vast majority of IR-RF studies have been performed on multi-grain aliquots, so the possible effects of variability of the IR-RF signal (e.g., differences in signal saturation or in proportions of RF emissions) from different grains has not received much attention in the literature so far, as detailed in section 2. Here, we investigate the IR-RF signal of five samples

from different locations at single-grain resolution and discuss the effect that the observed variability could have on multi-grain
aliquots. To assess whether the grain geochemistry influences the IR-RF signal and potentially the age limit of IR-RF, we
examined individual K-feldspar grains at the submicron resolution X-ray spectroscopy (SRX) beamline at the National
Synchrotron Light Source II (NSLS-II) at Brookhaven National Laboratory. Measurements at such a high-resolution may lead
to a better understanding of the luminescence kinetics in feldspars. We report on the feasibility and practicality of using μ-X-
ray fluorescence (μ-XRF) and μ-X-ray absorption near-edge fine structure (μ-XANES) techniques in investigating the IR-RF
signal origin and kinetics in K-feldspar.
**2 Method and rationale**
Identification of the defect type linked to the IR-RF signal and its concentration would enable us to better characterize the light
emission (signal sensitivity) in different types of feldspar, while identification of the origin of RF emissions could help us to
gain a better understanding about the apparent saturation or quenching of the IR-RF signal. μ-XRF and μ-XANES produce
high-resolution maps of elements and their oxidation states and are well suited for the purposes of our study. μ-XRF elemental
analyses are based on the characteristic fluorescence of atoms when stimulated with X-rays with a higher energy than their
ionization energy. In the case of μ-XANES, initial measurements of standards are run by varying the incident beam energy to
determine the specific energy equal to the absorption edge (binding energy of inner shell electrons) of the element or ion of
interest. This is apparent by an abrupt rise in the resulting fluorescence, which is different between oxidation states as they
require different minimum stimulation energies before ionization and subsequent fluorescence. μ-XRF maps using the obtained
absorption edge energies allow for maps of the different oxidation states of the same element.
The use of synchrotron μ-XRF allows us to improve the spatial resolution compared with a standard lab-bench μ-
XRF setup (e.g., Buylaert et al., 2018) by reducing the beam spot size from ~25 µm to 1 µm or 0.5 µm. Both the grain
geochemistry and crystallography should be investigated to characterize the defect type and its environment. In the present
study, we focussed only on geochemistry, though our results should be complemented with crystallographic studies in future
work.
The defect(s) responsible for the IR-RF emissions are still subject to debate. It has been suggested that IR-RF occurs
as a result of the change in the oxidation state of the participating lead (Pb) defect via the transition: $Pb^{2+} \rightarrow (Pb^+)^* \rightarrow Pb^+$
(Nagli and Dyachenko, 1986; Erfurt, 2003). A similar transition has been suggested for amazonite (see Ostrooumov, 2016),
but the direct connection between the Pb-centre and IR-RF has not yet been evidenced. Other reactions involving higher
oxidation states would also be possible but have not yet been observed or formally proposed. Additionally, the IR-RF signal
is composed of at least two separate emissions. Previous publications placed the main IR emission at 1.43 eV (865 nm) based
on Trautmann et al. (1999a, b) and Erfurt and Krbetschek (2003), but more recent work including corrections for the
spectrometer efficiency places the IR emission closer to 880 nm (Kumar et al., 2018; Riedesel et al., 2021; Sontag-González
et al., 2022). A second IR emission centred at 955 nm (1.30 eV) at lower intensity has also been identified (Kumar et al., 2018),
which partly overlaps with the 880 nm peak.
The presence of iron (Fe) in feldspar is known to lead to red RF (e.g., Telfer and Walker, 1978; Brooks et al., 2002;
Visocekas et al., 2014), with the maximum peak wavelength varying between 700 nm and 770 nm depending on feldspar
composition (Dütsch and Krbetschek, 1997; Krbetschek et al., 2002). Such observations are in line with the suggestion of more
than one component in the red photoluminescence of K-feldspar (Prasad and Jain, 2018). Despite the occurrence of the red RF
emission in $Fe^{3+}$ state, its initial state remains a subject of many debates, with conflicting opinions suggesting either $Fe^{2+}$ ($Fe^{2+}$
+ h → $Fe^{3+}$; here h stands for hole) or $Fe^{4+}$ ($Fe^{4+}$ + $e^-$ → $Fe^{3+}$; here $e^-$ stands for electron) (Kirsh and Townsend, 1988; Jain et
al., 2015). Recently, Kumar et al. (2020) argued that the initial state must be $Fe^{4+}$ based on their findings using
cathodoluminescence microscopy. Spectral analyses showed that, with dose exposure, the red RF emission (~710 nm emission
in K-feldspar) increases, while the 880 nm emission decreases (Krbetschek et al., 2000; Erfurt and Krbetschek, 2003; Kumar
et al., 2018; Frouin et al., 2019). The thermal stability of the ~710 nm emission has been, however, questioned (e.g., Krbetschek
et al., 2000). Such a reduced thermal stability might be an issue for IR-RF dating, as it has been suggested that the tail of the
~710 nm emission overlaps with the 880 nm emission, thus potentially playing a role in the shape of the measured IR-RF.
Such a contribution can be reduced to less than 5% of the IR-RF signal by using a bandpass filter centred at 850 nm (FWHM
40 nm), but can still affect the equivalent dose ($D_e$) value at doses near signal saturation (see Sontag-González and Fuchs,
2022). $D_e$ values are determined by sliding the IR-RF dose-response curve of grains containing the natural signal onto that
obtained after a full bleach of the same aliquot. In summary, although previous studies have identified factors that may
influence the IR-RF signal in several ways, e.g., whether the IR-RF signal originates from Pb, and is affected by the presence
of $Fe^{2+}$ or $Fe^{4+}$, a conclusive confirmation or comprehensive linkage between these factors is yet to be established.
A possible variability of the several RF emissions in individual grains has received scant attention so far. Trautmann
et al. (2000) were the first to analyse the IR-RF signal of individual K-feldspar grains. Using spectral measurements of 21 to
42 grains from three samples, they observed up to four emissions (IR, red, yellow, blue) with variable intensities (a fourth
sample appears in their figure 3 but is not mentioned in the main text). An IR-RF dose-response curve was only reported for
one grain, which had a similar shape, albeit a later onset of saturation, when compared to the response from the multi-grain
aliquot of the same sample. More recently, Mittelstraß and Kreutzer (2021) analysed 60 grains from two samples, of which
55% and 80% emitted a detectable signal. In that study, between one and three grains per sample (~9% of signal-emitting
grains for both samples) were rejected due to a bad match between the natural and regenerative curves, which might have been
caused by equipment issues, but also due to sensitivity changes (Varma et al., 2013). However, all grains that emitted a
detectable signal displayed the expected decay shape for IR-RF (decreasing signal with increasing dose). Likewise, own
laboratory observations indicated that the signal varies in sensitivity across feldspar minerals and can be contaminated for
various reasons, leading to spectral interference or quenching, ultimately influencing the saturation level and/or the shape of
the IR-RF signal (Frouin et al., 2017, 2019; Kumar et al., 2020).

To investigate these issues, first, we recorded IR-RF curves from individual grains in our luminescence dating laboratory at the Research Laboratory for Archaeology and the History of Art (RLAHA) at the University of Oxford (UK). Then, during our beam time (96 h), and as a proof of concept, we optimized the measurement conditions and obtained compositional maps of the individual K-feldspar grains. We paid particular attention to K, Ca, Fe and Pb. After analysing the μ-XRF maps, μ-XANES measurements were done at selective spots where Fe occurred in greater concentrations. Note: The atomic number of sodium (Na; another end member of the feldspar ternary system) is too low to be measured at the current SRX beamline.

## 3 Material and instrumentation

A total of five samples were selected to represent a diversity of i) geological context, ii) geochemistry, iii) shape of the IR-RF signal, and iv) age. Sample Gi326 from a Triassic sandstone from Bayreuth, Germany, is composed of 89% of K-feldspar (Sontag-González and Fuchs, 2022) and has previously been used as a reference sample in a laboratory comparison of IR-RF dating (Murari et al., 2021b). X7343 was collected from a Pliocene sediment at the Nyayanga site in the Homa peninsula, Kenya (Plummer et al., 2023). X7363 was taken from the Gele Tuff in the Turkana Basin, Kenya, dated by Ar/Ar at $1.315 \pm 0.002$ Ma (Phillips et al., 2023). Previous compositional analyses of Gele Tuff pumice feldspars (crushed clasts without density separation) indicate they are mostly composed of anorthoclase with smaller proportions of sanidine and plagioclase; K, Na and Ca contents ranged ~1–6 wt %, ~5–6 wt % and ~0–3 wt %, respectively, without appreciable differences between the grains' cores and rims (Phillips et al., 2023). Relatively high Ba contents of up to 0.8wt% were also reported in that study, with a positive correlation between Ba and Na contents. X7368 is a sediment sample collected above the Silbo Tuff ($0.751 \pm 0.022$ Ma, McDougall and Brown, 2006) and below the Kale Tuff (younger than the Silbo Tuff but not directly dated) in the Turkana Basin, Kenya. Sample H22550 is a coastal marine sample from Sula, Russia, dated by quartz single-aliquot-regenerative optically stimulated luminescence (OSL) at $103 \pm 8$ ka (Murray et al., 2007) and was used in the past as a reference sample to test the accuracy of IR-RF dating (Buylaert et al., 2012). All samples were prepared following conventional treatments (e.g., Preusser et al., 2008), including wet-sieving to isolate the desired grain size fraction, chemical treatment with HCl at 10% to remove carbonates and $H_2O_2$ at 30% for a few hours to a few days to remove organic matter, and density separation at 2.58 g cm$^{-3}$ using a heavy liquid solution to enrich K-feldspar grains. Sample H22550 was then etched with diluted HF (10%, 40 min). All grains were exposed under a solar simulator SOL Hönle 2 for a few days to reset their signal.

Unmeasured grains of sample X7343 were placed on a stub mount on a piece of carbon tape, then imaged with a scanning electron microscope (SEM) equipped for energy-dispersive X-ray spectroscopy (EDS) at Archéosciences Bordeaux (FR) (JEOL JSM-6460LV; detector: Oxford Instruments X-Max (51-XMX0002); software: Oxford Instruments INCA version 4.11). The SEM was operated at 20 kV and 55 μA beam current. Sample X7343 is referred to as BDX22338 in the Archéosciences Bordeaux system.

IR-RF measurements were recorded with a *lexsyg research* luminescence reader fitted with an annular $^{90}Sr/^{90}Y$ beta source (Richter et al., 2013) using a bandpass filter centred at 850 nm (FWHM 40 nm) mounted in front of a Hamamatsu H7421-50 photo-multiplier tube. Measurements were performed at 70°C, following Frouin et al. (2017). Multi-grains and single-grains were measured on stainless steel cups. High-resolution compositional analysis of the grains was undertaken at the SRX beamline at NSLS-II (Chen-Wiegart et al., 2016). After IR-RF measurements, the grains were removed from the stainless steel cups and fixed on a polymer microscope slide (UVT acrylic; Agar Scientific) with a small piece of carbon tape to avoid misplacement during measurement (supplementary Fig. S1). μ-XRF maps were obtained by scanning across pre-selected regions on the grains with low topographic changes (90 x 90 μm maps, with a step size of 0.67 μm and an integration time of 0.1 s). The incident X-ray beam was focussed by a pair of Kirkpatrick-Baez mirrors. An incident beam energy of 13.5 keV was used for the μ-XRF measurements. The excited elements' characteristic fluorescence was obtained from the sum of the four elements of a silicon drift detector. All μ-XRF measurements were normalized to the corresponding incident X-ray flux ($I_0$) (supplementary Fig. S2).

The μ-XANES maps cover 60 x 60 μm in steps of 0.5 μm, thus creating a grid with 120 x 120 data points (i.e., 14 400 μ-XRF spectra). To obtain maps of Fe-states in our samples, we varied the incident beam energy according to the absorption edge values obtained from the μ-XANES spectral measurements of Fe standards (Fe foil, pyrite, hematite). The μ-XANES maps were measured three times to obtain μ-XRF emission spectra restricting the Fe species to either (i) the total Fe (at 7.275 keV), (ii) the sum of $Fe^{3+}$ and $Fe^{2+}$ (at 7.134 keV), and (iii) only from $Fe^{2+}$ (at 7.122 keV). The difference between the intensity levels of the latter two measurements can qualitatively give the intensity levels of $Fe^{3+}$, i.e., $I_{Fe2+ \& Fe3+} - I_{Fe2+} = I_{Fe3+}$ where $I$ (a.u.) refers to intensity, thus, the μ-XANES map of $Fe^{3+}$. We also attempted to record Pb states, however, the Pb standard available at the SRX beamline was fully oxidized, which hindered establishing the correct beam energy for mapping. Therefore, no Pb oxidation state maps were possible. μ-XRF and μ-XANES data were analysed using the open-source software PyXRF v1.0.23 (Li et al., 2017) and ATHENA v0.9.26 (Ravel and Newville, 2005), respectively. Maps and plots were created using **R** (R Core Team, 2022).

## 4 Results

### 4.1 Multi-grain IR-RF signal

The IR-RF signal of a multi-grain aliquot of 8 mm diameter of sample X7343 was first measured. The aliquot contained hundreds of grains. The expected IR-RF signal of K-feldspar grains is a decaying function, e.g., a stretched single-exponential (Erfurt et al., 2003). For sample X7343, however, we observed an unexpected shape of the IR-RF, consisting of a signal decrease until 500 Gy succeeded by an increase, roughly following a saturating exponential shape that keeps increasing beyond ~3800 Gy. The regenerative signal for one representative aliquot is shown in figure 1 (top right). We hypothesized that the unexpected signal increase originates from a different source, potentially from a coating around the grains due to the observation of a pinkish/reddish hue on some grains. Clay, Fe-oxide or other grain coatings are a common occurrence and

additional preparation steps are sometimes undertaken to remove them prior to luminescence measurements (e.g.,
Jayangondaperumal et al., 2012; Lomax et al., 2007; Rasmussen et al., 2023). We attempted to remove this signal
contamination using different chemical treatments such as HF, regal water, and heated regal water, however, without success.
This suggests the signal is not originating from a coating. Therefore, we decided to investigate the mineral composition of
sample X7343, using SEM-EDS on 118 randomly selected grains. Despite using density separation to isolate K-feldspar grains
during chemical pre-treatment, we found that this sample was mainly composed of low-K grains (Fig. 1; top left). Indeed, over
half of the grains had K-contents less than 2% and less than 5% of the grains had K-contents above 11%. The remainder
exhibited K-contents between 2% and 10%. Note that a K-feldspar end member is 14% K (e.g., Gupta, 2015). The low-K
grains, which correspond to the majority of grains, also had high Fe-contents of ~10%.
We then tested whether it was possible to isolate the desired decreasing IR-RF signal by handpicking grains based on
their visual appearance under a microscope. Between 10 to 30 grains were placed onto two aliquots, one for transparent shiny
angular grains and one for white-pinkish rounded grains. The regenerated IR-RF signals showed a clear distinction between
the two aliquots (Fig. 1), proving it is possible to separate the two observed IR-RF shapes.
By manually selecting the grains based on their shape and colour and the subsequent multi-grain IR-RF
measurements, we made three important observations: i) The decreasing IR-RF signal originates from a small number of grains
(less than 5%), presumed to be K-feldspar. ii) The IR-RF signal of these grains decreased beyond 3800 Gy without reaching a
plateau, indicating that a dose could be estimated beyond that value. iii) The increasing IR-RF signal originates from a different
subset of grains, presumed to be the low-K, Fe-rich minerals identified via SEM-EDS.

**4.2 Single-grain IR-RF characterisation**

To further investigate this phenomenon, we measured the IR-RF signal of 22 individual grains from five samples of different
origins (between one to eight grains per sample, Table 1). Each grain was manually placed on a sample holder (cup) and their
signal was recorded over a 3265 Gy beta irradiation. For each grain, their IR-RF signal shape falls into three categories (Fig.
2): Category #1 for grains with a decreasing IR-RF signal, category #2 for grains with an increasing IR-RF signal, and category
#3 for grains with a flat signal. Within categories #1 and #2, the saturation level of the individual grains varies (Fig. 2b, d).
Among the five samples, one is a tuff, and two are originated from nearby volcanic environments and might, thus, be
expected to yield abnormal luminescence behaviour. Common issues with volcanic samples are dim signals, different
proportions of emissions, high fading rates and complex grain mineralogy (e.g., Krbetschek et al., 1997; Guérin and Visocekas,
2015; Joordens et al., 2015; Sontag-González et al., 2021; O'Gorman et al., 2021). However, we also observed the unwanted
increasing IR-RF signal for one of the four grains for sample H22550, which is from a coastal sand deposit. The significance
of this find is illustrated in figure 3, where curves representing the signals of individual grains from categories #1 and #2 were
added together to simulate a multi-grain aliquot. We used the curves obtained from fitting a single stretched exponential decay
function to the normalized data of one grain of sample X7343 (category #2) and one of Gi326 (category #1), since no category
#1 grain was measured for sample X7343. When the total signal of the theoretical aliquot was composed of more than 50% of
signal from the category #2 grain, we observed the same decay shape as in figure 1 for a multi-grain aliquot of sample X7343.
Importantly, a synthetic mixture containing 20% of grains from category #2 still displayed the decaying shape characteristic
of category #1 grains. However, the curvature of its dose-response curve was altered, i.e., saturating earlier than the 'pure'
grain. Possible differences in long-term signal stability between the two grain categories could cause differences between the
summed curves of natural and regenerated IR-RF signals and thus lead to inaccurate equivalent doses for these mixtures.
Further, our results demonstrate that a satisfying IR-RF signal can be measured for all our samples, but only by
selecting grains with the appropriate IR-RF characteristics (presumably K-feldspar grains). We hypothesize that the low-K
grains with high Fe-content are the source of a contaminant IR-RF emission, which if not removed might result in a wrong
equivalent dose estimation (i.e., a wrong age estimate).

## 4.3 Sub-single grain µ-XRF elemental maps

Utilising µ-XRF, we identified up to 15 elements in the grains (see supplementary figure 3 for the total µ-XRF spectra). We
then fitted each of the spectra in the 135 by 135 pixel grid (i.e., 18 225 spectra) for each grain to obtain maps describing the
XRF intensity of each identified element. These maps only serve as qualitative indicators for the presence of elements and are
not corrected for the element-specific emission intensity or the energy-dependent efficiency of the detectors. For two grains,
we recorded additional µ-XRF maps to characterize visible inclusions (see table 1).
First, we consider only the presence/absence of each element with the IR-RF signals previously obtained. Most grains
across all categories contain K, Pb, and Fe, among other elements. Among the grains displaying a decreasing IR-RF signal
(category #1), all contain Ba (Fig. 4, middle), which is less present in grains from categories #2 and #3. Further, most grains
from categories #2 and #3 contain Ca, Ti and Mn, which are rare in the grains from category #1. The µ-XRF intensities also
allow for a qualitative comparison of elemental composition. As shown in the boxplots in figure 4 (right-hand side), category
#1 grains differ from those in category #2 and #3 primarily by a higher µ-XRF signal contribution from K and a lower
contribution from Fe.
If we compare the relative intensities of Pb, Fe and K, we can identify a pattern in the composition of grains from
each category (Fig. 5). Grains from category #1 tend to have high proportions of K and Pb and medium-to-low proportions of
Fe. All grains from categories #2 and #3 have medium-to-high proportions of Fe, and most have low levels of K and Pb. No
grains from categories #2 or #3 have high levels of both K and Pb. The element that distinguishes grains from categories #2
and #3 is Ca, which is only present in category #2 grains to a high proportion (see also supplementary figure 3).
Interestingly, the grains from category #3 cluster relatively close to those from category #2, suggesting that the
elemental composition alone is not responsible for the lack of IR-RF signal but that the spatial configuration of the elements
is another important factor to consider. This is exemplified in figure 6, which contains elemental maps of K, Pb and Fe for one
grain of each category. The category #1 grain (top row) tends to have K and Pb co-localized (overlap shown in green), whereas
the category #3 grain (bottom row) also contains both elements, but they appear in separate locations (shown in cyan and
yellow). In this grain, K appears co-localized with Fe (overlap shown in dark blue). Our preliminary observations require
broader confirmation, but they are in line with the current hypotheses of emission origins for the K-feldspar IR-RF signal
decreasing with dose (due to Pb) and a contaminating red RF signal increasing with dose (due to Fe). Furthermore, the lack of
an IR-RF signal in category #3 grains appears to stem from low levels of K co-existing with high proportions of Fe, supporting
the observations made by Kumar et al. (2020).

**4.4 Mapping oxidation states with µ-XANES**

By analysing absorption of X-rays near the absorption edge, µ-XANES spectra can provide information on the presence of
potential oxidation states of an element, as shown in figure 7 through measurements of different standards of Fe; an increase
in the oxidation state is generally accompanied by a shift in the absorption edge to higher energy (Fig. 7, inset).
We targeted the Fe-rich region of a category #1 grain (X7368) for mapping (location shown by the blue square in
figure 6). Figure 8 shows the µ-XANES maps of three oxidation states ($Fe^{3+}$ and $Fe^{2+}$ combined, $Fe^{2+}$, and $Fe^{3+}$), all normalized
to the maximum intensity level of the total Fe map. These maps suggest that Fe exists in the top section of this feldspar grain
in its $Fe^{3+}$ and $Fe^{2+}$ states. Note how $Fe^{2+}$ is mainly clustered in one region, possibly within a mineral inclusion with a rim of
$Fe^{3+}$.

**5 Conclusions and future work**

We demonstrated that individual K-feldspar grains of the same five samples display different IR-RF behaviour, illustrated by
different signal decay shapes (i.e., increasing or decreasing with dose and different saturation levels). These behaviours are
cumulative (see figure 1), and therefore, the IR-RF signal of a multi-grain aliquot can lead to inaccurate equivalent doses.
Despite the use of chemical preparations to remove contaminants, manually picking individual grains was necessary to isolate
K-feldspar grains, which is unrealistic for routine dating applications in a low-light laboratory. A more realistic way to remove
such contamination is by selecting K-feldspar grain populations by isolating the emission signal of individual grains with an
imaging system. Here, we wanted to gain a further understanding of the production and origin of the emission signal, which
ultimately will help us design a more appropriate imaging system for IR-RF dating. For sample X7343, we show through
SEM-EDS analyses that the different emissions can be linked to different grain mineralogy. Since K-feldspar grains are known
to be heterogeneous on a subgrain level, we propose synchrotron-based X-ray spectroscopy to characterize the grains on a
submicron scale and investigate the origin of the IR-RF and other linked emissions. Information on the oxidation states of,
e.g., Fe possibly allows for the characterisation of the reactions behind the electronic changes leading to radiofluorescence.
In the preliminary work presented here, we successfully applied µ-XRF and µ-XANES at the SRX beamline (NSLS-
II) to obtain element and oxidation state maps of regions of interest within individual K-feldspar coarse grains previously used
for IR-RF measurements. We were able to correlate the desired IR-RF signal shape (category #1) with compositions of high
proportions of K, Pb, and Ba and low proportions of Fe. High proportions of Fe in the µ-XRF spectra were found in grains of
categories #2 and #3, but the possible role of Fe as a contaminant remains unclear. During our next beam time, we will polish

the grains down to a uniform surface prior to μ-XRF and μ-XANES measurements to limit surface effects. Such a setup will also allow us to directly test the hypothesis that the contaminating IR-RF signal is coming from an element present at the surface of the grain (e.g., iron coating possibly due to weathering), but not within the grain.

The relation between the chemical composition, crystal structure, and the shape of the IR-RF signal in individual K-feldspar grains is still poorly understood, and efforts should be made to identify and quantify at high resolution the element responsible for producing the IR-RF signal with the highest dynamic range (i.e., saturation at high dose). Our future work will include implementing a second detector to simultaneously measure μ-XRF/μ-XANES and the IR-RF signal induced by the X-rays. Though not widely used, X-rays are a suitable alternative to radioactive sources for luminescence dosimetry including RF. The dual detection will allow us to isolate emissions from different mineral inclusions and directly correlate them to the elemental composition, thereby assessing the extent of overlap of the desired IR-RF emission centred at 880 nm and contaminating ones such as the possible unstable red emission associated with $Fe^{3+}$.

**Data availability**

The SEM dataset and the original data used to produce μ-XRF maps are available online (Sontag-González et al., 2023).

**Author contribution**

MF, JT, RK and JLS designed the experiments and prepared the samples. RK, JLS, and MF carried out the IR-RF measurements. SK organized and analysed the SEM EDS measurements. MF and JT carried out the μ-XRF and μ-XANES measurements. RK, MSG and MF analysed the results. MSG and RK prepared the manuscript with contributions from all authors. MF, JT and JLS obtained funding.

**Competing interests**

The authors declare that they have no conflict of interest.

**Acknowledgements**

We are grateful to Sumiko Tsukamoto, Svenja Riedesel and an anonymous referee for very constructive comments on earlier versions of this manuscript. We thank Yannick Lefrais for operating the EDS at Archéosciences Bordeaux (former IRAMAT-CRP2A) in 2018. This research used the SRX beamline of the National Synchrotron Light Source II, a U.S. Department of

Energy (DOE) Office of Science User Facility operated for the DOE Office of Science by Brookhaven National Laboratory under Contract No. DE-SC0012704.

**Financial support**

This work was supported by the Natural Environment Research Council (grant number NE/T001313/1); and a Stony Brook University-Brookhaven National Laboratory Seed Grant (#94508). The SEM analysis at Archéosciences Bordeaux was supported by the Agence Nationale de la Recherche (grant no. ANR-10-LABX-52).

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

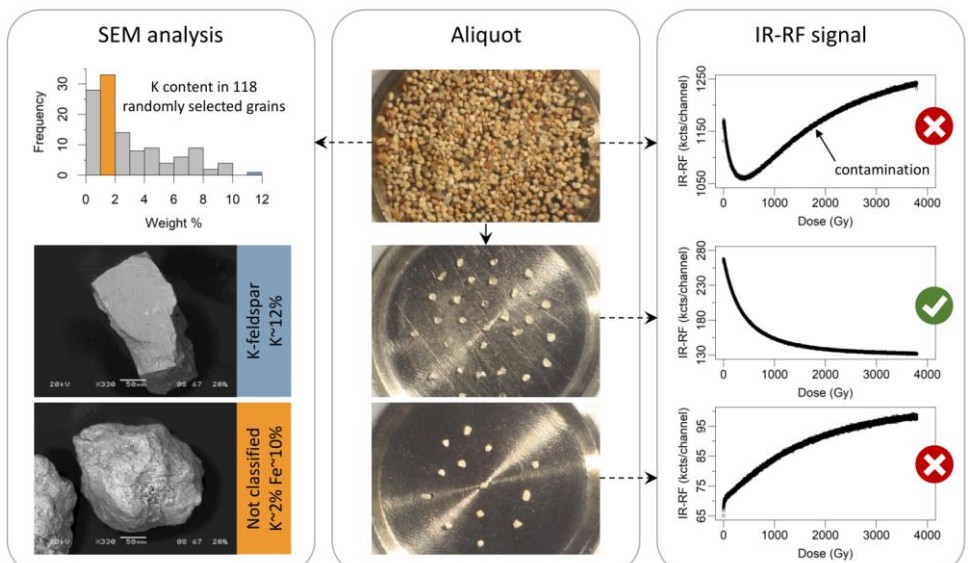


**Figure 1:** Illustration showing how contamination of the IR-RF signal can be removed by selecting only K-feldspar grains from sample X7343. The regenerative IR-RF curves were obtained from aliquots containing hundreds of unsorted grains (top) or 10–30 grains manually sorted into transparent shiny angular grains (middle) or white-pinkish rounded grains (bottom). The histogram shows the K-content determined by SEM-EDS for 118 grains (not measured for IR-RF). Representative examples of grains classified as K-rich and contaminating Fe-rich grains are shown.

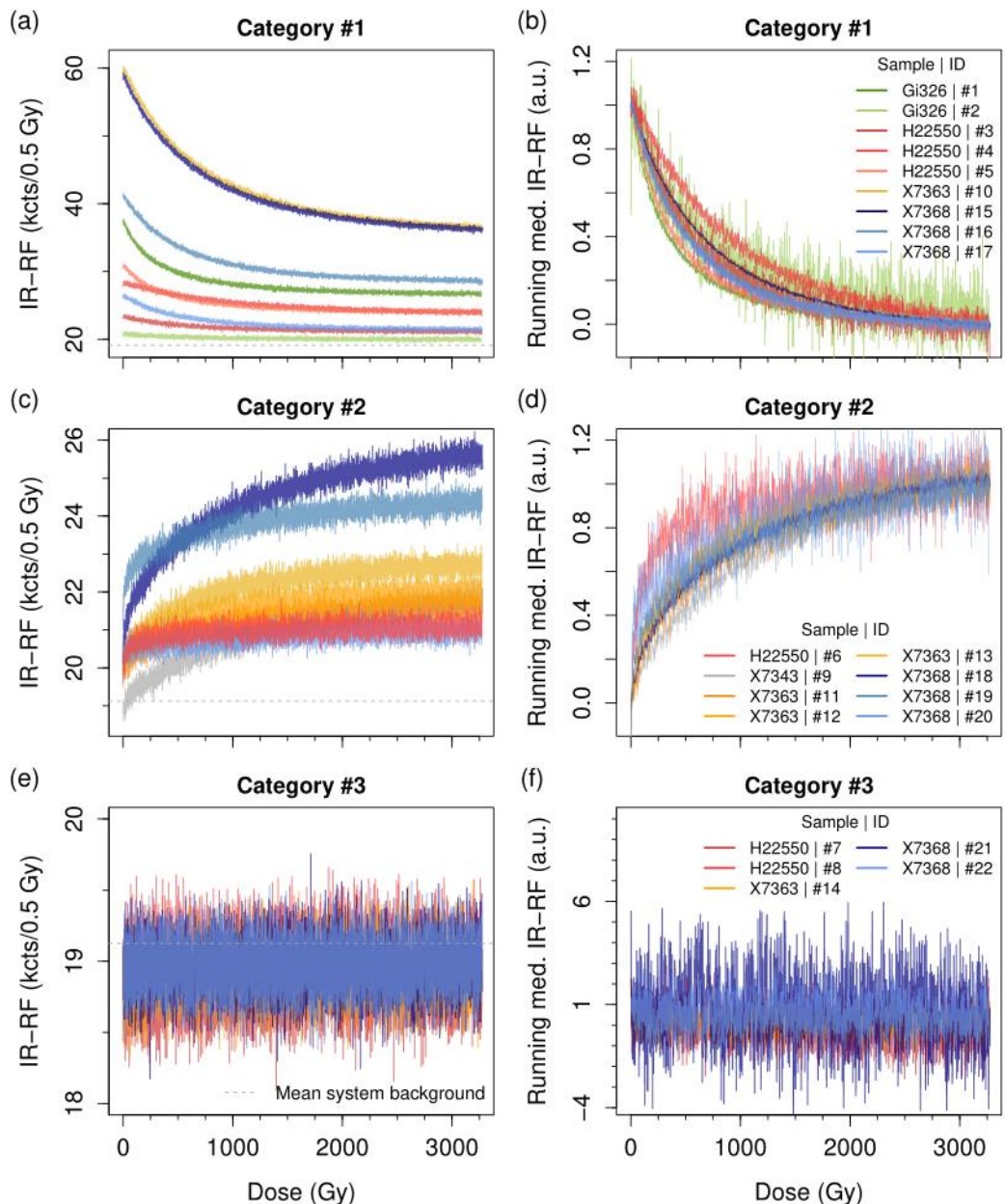

467

**Figure 2:** IR-RF dose-response curves of individual grains obtained after bleaching. Categories #1–#3 refer to grains with decreasing, increasing or no detectable signal, respectively (one category per row). The curves are shown (a, c, e) unnormalized and without background correction and (b, d, f) with intensities normalized to the signal maxima (defined as the median value of (b, f) the initial and (d) the final 20 channels) after subtracting as background the minimum signal of each grain (defined as the median value of (b, f) the final and (d) the initial 20 channels). For better visualisation, the normalized plots show the running median IR-RF with a window of 7 values. The system background was determined as the mean value obtained from measuring 5 empty cups under the same conditions as the grains.

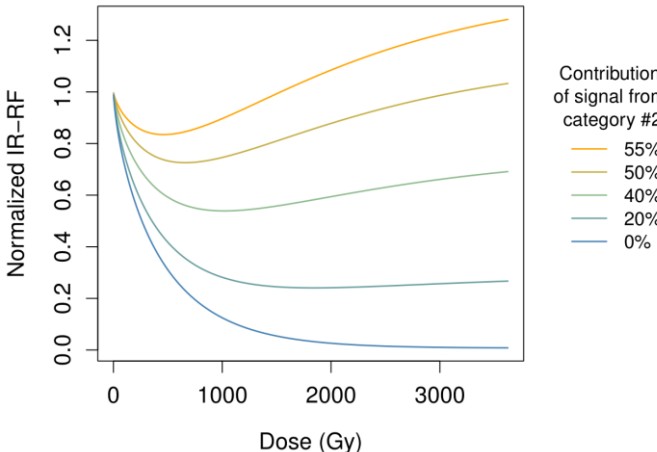


**Figure 3:** Simulated dose-response curves of theoretical aliquots varying the proportion of grains from categories #1 (desired decreasing
signal) and #2 (increasing signal). The curves are the sum of two stretched exponentials using parameters obtained from fits of grains from
samples Gi326 (category #1) and X3743 (category #2). The higher the signal contribution from category #2 grains, the more aberrant the
sum curve becomes.

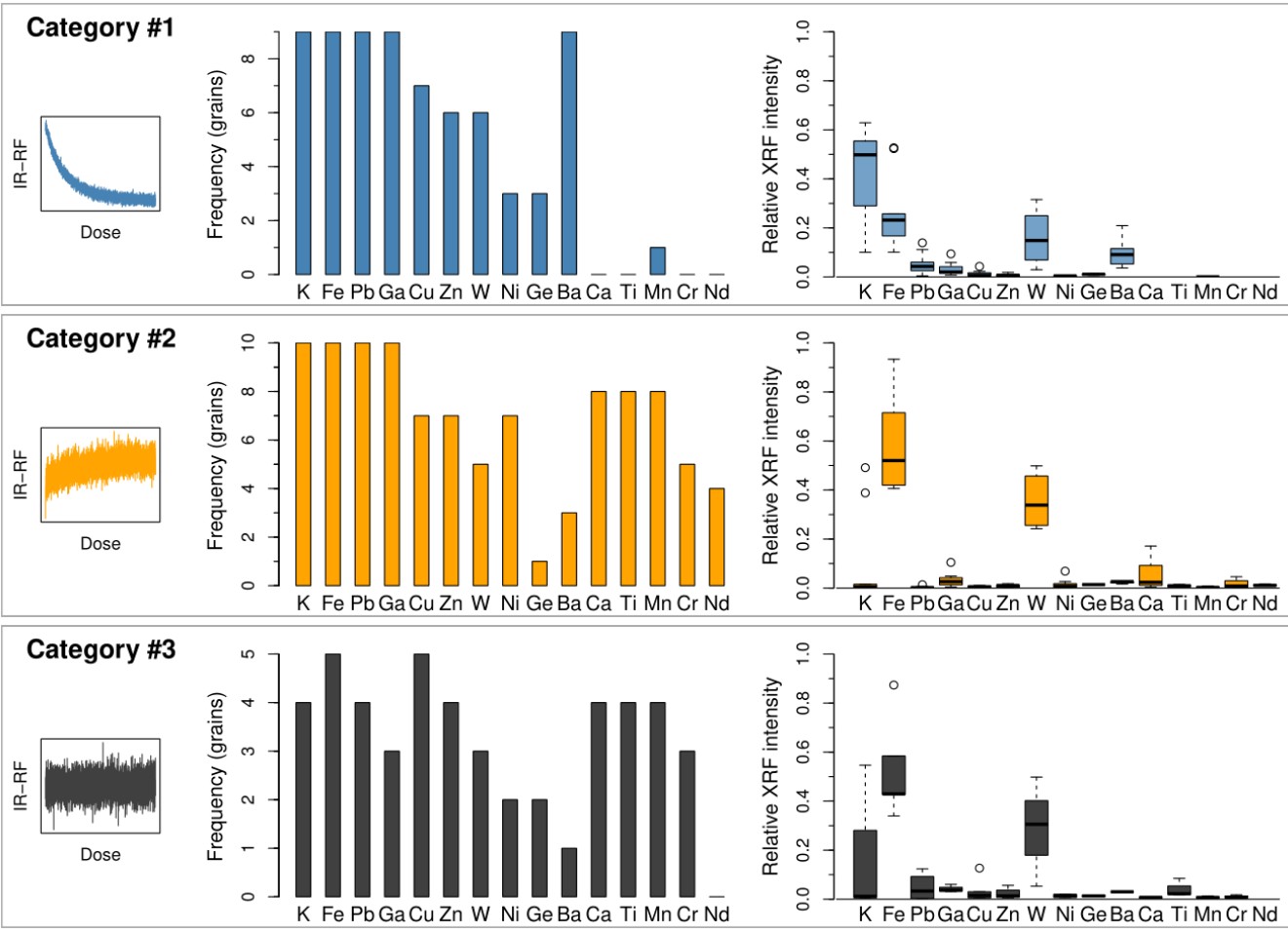


**Figure 4:** Bar charts of elements identified in µ-XRF spectra and boxplots of the relative µ-XRF intensities for grains in three categories, as exemplified in the insets: decreasing IR-RF signal (category #1), increasing IR-RF signal (category #2) or flat IR-RF signal indistinguishable from the background (category #3) during beta irradiation.

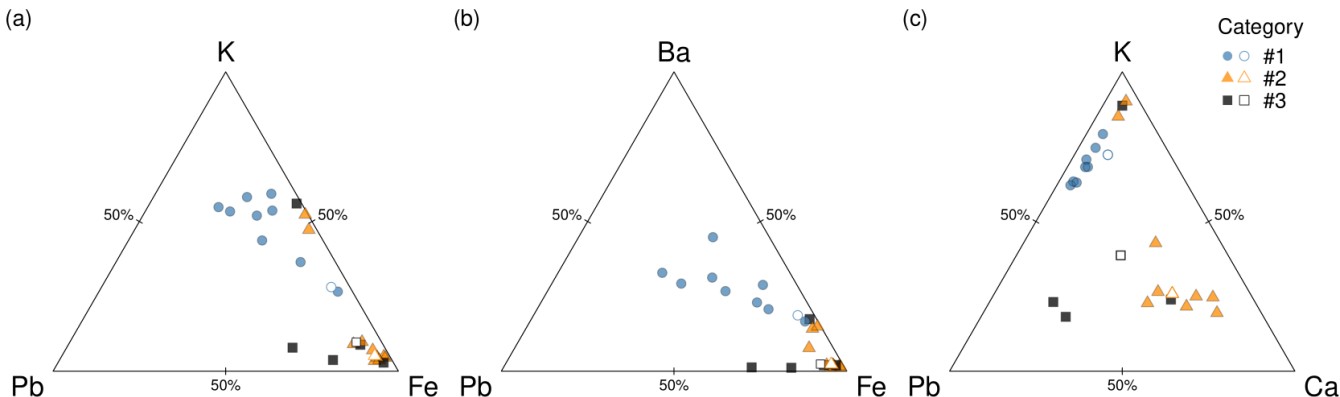

**Figure 5:** Ternary diagrams of relative μ-XRF intensities attributed to (a) K, Fe and Pb, (b) Ba, Fe and Pb, and (c) K, Ca and Pb for grains of the three categories. Note that the contributions are not calibrated to mass or stoichiometry. The relative K contribution is, thus, not directly comparable to the K-feldspar K-content. The three grains shown in figure 6 are marked as open symbols in each ternary diagram.

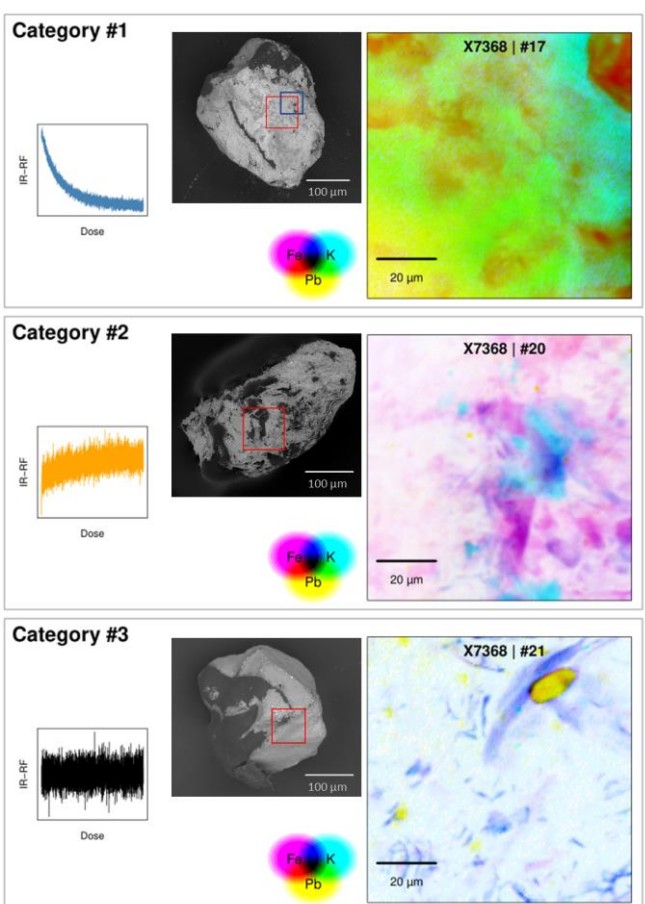

489

**Figure 6:** Illustration showing three IR-RF curves obtained from three grains of sample X7368, classified as follows: decreasing IR-RF signal (category #1), increasing IR-RF signal (category #2) or flat IR-RF signal indistinguishable from the background (category #3) during beta irradiation. μ-XRF spectra were measured from the area bordered by red squares on the SEM images of the grains. The maps show the presence of K, Fe, and Pb on the same grains as the IR-RF curves. The elemental compositions are shown overlaid, with the colour scales normalized to the maximum contribution of each element for each grain. The area bordered by a blue square in the category #1 grain corresponds to the map shown in figure 8.

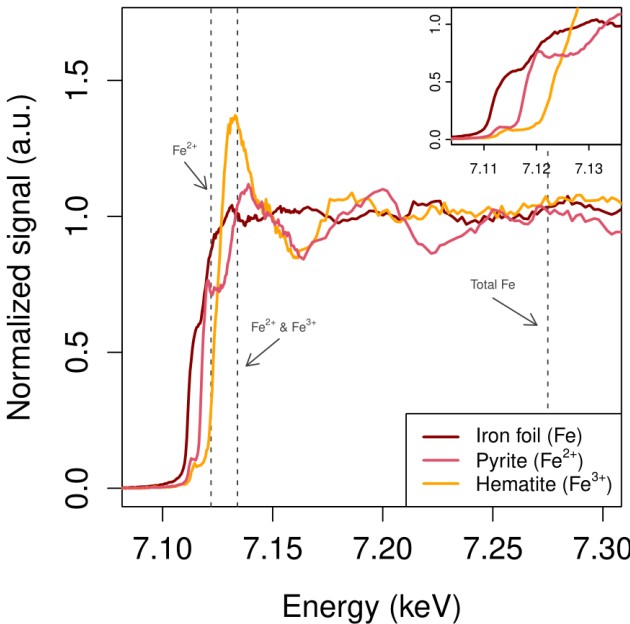

**Figure 7:** μ-XANES spectra of Fe standards. The dashed vertical lines indicate the incident beam energies necessary to isolate emissions
from specific oxidation states. The inset shows a magnification of the energy region relevant to determine the incident beam energies.


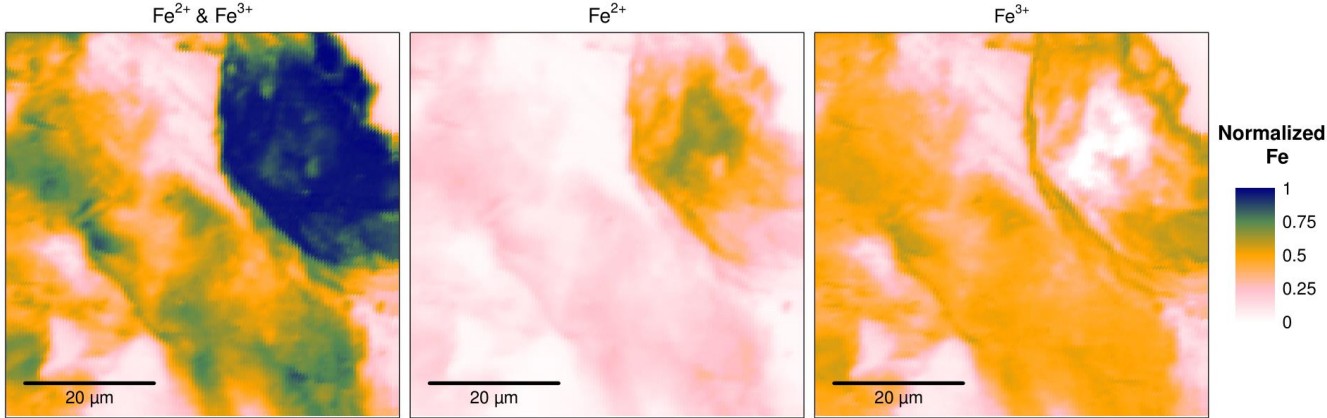

**Figure 8:** Maps of Fe oxidation states for a grain of sample X7368 (category #1; ID #17). Intensities are normalized to the maximum
intensity of total Fe.
**Table1:** Overview of measured grains. Categories #1–#3 refer to grains with decreasing, increasing or no detectable signal, respectively.
For two grains, two regions each were mapped, so we measured a total of 24 µ-XRF maps.

| Sample | Grain size (µm) | Number of measured grains | | |
|--------|-----------------|-------------|-------------|-------------|
| | | Category #1 | Category #2 | Category #3 |
| Gi326 | 90–200 | 2 | 0 | 0 |
| H22550 | 180–250 | 3 | 1* | 2 |
| X7343 | 180–255 | 0 | 1* | 0 |
| X7363 | 180–255 | 1 | 3 | 1 |
| X7368 | 180–255 | 3 | 3 | 2 |
| Total | | 9 | 8 | 5 |

*For these grains, two regions were mapped by µ-XRF: the grain 'matrix' and an inclusion.