# Peer review of "Short communication: Synchrotron-based elemental mapping of"

_Geochronology, 2023_

## Referee Comment (RC2)

**Review of "Synchrotron-based elemental mapping of single grains to investigate variable infrared-radiofluorescence emissions for luminescence dating"**

This is an interesting work in which authors tried to identify the causes in variation in shapes of the IR-RF curves and attributed them to variations in elemental concentrations inside mineral grains. The authors also conclude such variations can result in wrong estimation of doses. Although methodology adopted seems reasonable, there are still several scientific aspects, which are not clear and need to be addressed.

**Comments:**

1. Page 2 line 38, Introduction: "rely on the capacity of". Its not capacity, its property of defects. Change appropriately.
2. Page 2 line 40, 41, Introduction: "In the laboratory, the total amount of energy stored" and "energy absorption rate (dose rate, Gy a-1)". Energy per unit mass is dose.
3. Page 2 line 43, Introduction: "quartz because of its high abundance and resistance to weathering". Besides these two, the fast to bleach OSL signal makes it most appropriate for geological dating.
4. Page 2 line 45, Introduction: "considering a low dose rate of 1 Gy ka-1"
5. Why to mention 'low' here? Please delete low
6. Page 2 line 64, Introduction: "saturation cap at around 1500 Gy, reducing" cap can be deleted
7. Page 3 line 80, Method rationale: "study, between one and three grains (~…….. curves" Statement not clear, consider revising.
   What is the reason for bad match? Bad match is often observed due to sensitivity changes. Pls refer (Varma, V., Biswas, R.H., Singhvi, A.K., 2013. Aspects of Infrared Radioluminescence dosimetry in K-feldspar. Geochronometria 40, 266-273.)
8. Page 3 line 83, Method rationale: "grains that emitted a detectable signal displayed the expected decay shape." What is meaning of expected shape here? exponential decay?
9. Page 3 line 89, Method rationale: "With dose exposure, the 955 nm emission increases and overlaps with the 880 nm peak." Why is it so? How dose increases the 955 nm emission, does it mean more 955nm recombination centres are being regenerated? Does it reflect multiple trap system?
10. Page 4 line 101, Method rationale: "The thermal stability of the ~710 nm emission has been, however…. the measured IR-RF" The red TL emission in feldspar is generally considered more stable than conventional IRSL method, why the red IR-RF is unstable?
11. Page 4 line 111, Method rationale: "The μ-XRF and μ-XANES techniques are best suited for this purpose by producing high-resolution maps of elements and their oxidation states".
    It will be good to provide some details about the mentioned techniques and their usefulness for present work.
12. Page 3 line 83, Method rationale: "The use of synchrotron μXRF allows us to improve the spatial resolution compared with previous uses of μXRF (e.g., Buylaert et al., 2018) by reducing the beam spot size from ~25 μm to 1 or 0.5 μm"
    It is indeed impressive that spot size is smaller and we can work at higher resolutions, but how will it effect S/N ratio and thus elemental concentration estimation? In addition, since spot size is smaller, only few grains analysis may be possible. In such cases, how can we get the statistical representation of entire grain population just based on few grains studies?
13. Page 4 line 128, Material and instrumentation:

Normally Tuff samples are expected to contain Fe rich species. Is this a deliberate choice to look the effect of Fe in the samples as 2 out of 5 are tuff samples?

14. Page 5 line 134, Material and instrumentation: Why only the sample H22550 was etched with HF. Why not same is performed for other samples?

15. Page 5 line 134, Material and instrumentation: "Multi-grain and single-grains … National Laboratory"
How correspondence between IRRF and XRF signals is established?

16. Page 5 line 143, Material and instrumentation: "2016). Grains were fixed on a polymer microscope slide…………" What are spectral and luminescence characteristics of the base material used?

17. Page 5 line 144, Material and instrumentation: "XRF maps were obtained by scanning across pre-selected regions on the grains 90 x 90 µm maps," What is the basis of ROIs selection?

18. Page 5 line 149, Material and instrumentation: "resolution of 0.67 µm was achieved by focusing the beam with" Is it Xray beam focussing or luminescence focussing, please specify.

19. Page 5 line 150, Material and instrumentation: " An incident beam energy of 13.5 keV was" Why this energy chosen any specific reason?

20. Page 5 line 150, Material and instrumentation: "fluorescence was detected through the sum of 4 silicon drift detectors" Why these four detectors were used? Can we use PMT instead? whats the advantage we get with use of these detectors.
Can you provide geometry of measurements and experimental setup?

21. Page 5 line 152, Material and instrumentation: "All XRF measurements were normalised to the corresponding incident X-ray flux " X-ray sources are normally found inhomogenous spatially and temporally. Does this can effect your measurements?

22. Page 5 line 150, Material and instrumentation: "The XANES maps had a resolution of 0.5 µm (60 x 60µm)." Are units correct? How does 60 um X 60 um translate to 0.5um? not clear.

23. Page 5 line 150, Material and instrumentation: "we varied the incident beam energy according to the absorption edge values obtained from the µXANES measurements of Fe standards (Fe foil, pyrite, hematite)." How the specific absorption edge values were estimated?

24. Page 5 line 150, Material and instrumentation: "(i) the total Fe (at 7.275 keV)," The energies mentioned here are quite precise. How much is normally the resolution. Since electronic energy levels of specific elements are quite low in energy (~few eVs) compared to what is being provided, then why this much precision is needed?

25. Page 6 line 165, Results: "500 Gy succeeded by an increase, roughly following a saturating exponential shape" What is reason behind increase to saturating exponential behaviour? Why should there be an increase at all considering physics aspect? What is the  nature of sample X7343, Is it similar to volcanic tuff?

26. Page 6 line 169, Results: "contamination, potentially coming from coating around the grains, we" Why is it assumed that coating could be responsible?

27. Page 6 line 171, Results: "Despite using density … high Fe content". Does that mean it is Na or Ca feldspar grains? Have you performed XRD analysis on bulk to see the mineralogy of samples?

28. Page 6 line 176, Results: "their visual appearance under a microscope" What were the visual indicators considered for choosing K-Feldspar?

29. Page 6 line 178, Results: "grains. The regenerated IR-RF signals showed a clear distinction between the two aliquots (Fig. 1), proving it is possible to separate the two observed IR-RF shapes." This is quite a qualitative way. I am not sure how to progress using only visually inspected grains. The  visual appearance and selection can vary depending upon geological

settings of grains as well as person observing them. Is there any other rigorous way of making such selection?

30. Page 6 line 184, Results: "presumed to be the low-K, Fe rich minerals identified via SEM-EDS" Low K means possibly high Na or Ca, why only Fe is considered. Fe if present should be in form of defects, which should be in ppm level. Can uXRF measure to such low concentration levels? If Fe is appearing as major element in feldspar separates, it means it is present in stoichiometric formula and in that case, mineralogy of sample would be different. Please suggest if I am missing something.

31. Page 6 line 186, SG IR-RF characterisation: "signal of twenty-two individual grains coming" Can you specify mineralogy of each grains, which are picked for such measurements?

32. Page 6 line 165, SG IR-RF characterisation: "2): Category #1 for grains with a decreasing IR-RF signal, category #2 for grains with an increasing IR190  RF signal, and category #3 for grains with a flat signal". How many grains falls in each category and is there any link to the provenance.

33. Page 6 line 165, SG IR-RF characterisation: "we also observed the unwanted decreasing IR-RF signal for one of the four grains for sample H22550, which is from a coastal sand deposit." We expect IR-RF signal to decrease with irradiation, so why it is said unwanted ?

34. Page 6 line 193, SG IR-RF characterisation: "When the total signal of the theoretical aliquot was composed of more than 50% of signal from the category #2 grain, we observed the same decay shape as in figure 1 for a multi-grain aliquot sample X7343"
Obviously, since the two different category of grains having two different IR-RF characteristics are being added, so the result will depend on the proportion of the individual populations in the mixture. More importantly, it is important to know, how these two grains are different with respect to crystallography or stoichiometry or defects concentration. Is the nature of curve repeatable over repeated bleaching and irradiation cycles?

35. Page 7 line 201, SG IR-RF characterisation: As mentioned by authors, long-term signal stability may not be there for bad traps, is there a way to prove it? How do we know it without experiment?

36. Page 7 line 204, SG IR-RF characterisation: "Further, our results demonstrate…" I agree with this statement, but it is still not clear how can we segregate K-Feldspar and other minerals. Manually it will not be possible on routine basis.

37. Page 7 line 207, Subgrain μXRF elemental maps: "We then fitted each of the **18 225** spectra for…."  this statement is not clear

38. Page 7 line 212, Subgrain μXRF elemental maps: "characterise visible inclusions (see Table 1)." The number of grains analysed per samples are quite small to represent the statistics of system.  Can we consider them as representative of whole samples? It is difficult to conclude unless sufficient data points exists.

39. Page 7 line 214, Subgrain μXRF elemental maps: "all contain K, Pb, Fe and Ba, among other elements (Fig. 4)." What is typical concentration of these elements? Considering K is a major element present in stoichiometric formula, how much is relative concentration of the other elements?

40. Page 7 line 223, Subgrain μXRF elemental maps: "grains from category #3 cluster relatively close to those from category #2, suggesting"
How and why does this effect the IR-RF properties? These are observations, but what is the reason for IR-RF signal due to such clusters is not clear.

41. Page 7 line 237, Mapping oxidation states with μXANES: "suggest that Fe exists on the surface of this feldspar grain in its $Fe^{3+}$ and $Fe^{2+}$ states." It is great that using uXANES, we could map

the presence of Fe on the surface of feldspar grain, but luminescence or IR-RF is a volumetric phenomenon. How this observation is helpful in explaining the IR-RF signal.

---

## Author Response (AR1)

**Authors' response to Svenja Riedesel's general comments:**

Thank you for the comments and feedback. Below we are responding to each of your comments.

General comment #1: We agree that more data would automatically make our observations more robust. However, getting granted access to the National Synchrotron Light Source II for an experiment is highly competitive. All beam time is allocated based on a peer-reviewed proposal process. We are continuously submitting proposals to the Brookhaven National Laboratory with the hope of being awarded beam time in the near future.

In our opinion, the sample preparation and methodology we have implemented during this beam time has never been used in the luminescence dating discipline. Because of its pioneering aspect, it should be worth publishing as a short communication. In addition, our current dataset and results are advancing our understanding of oxidation states in feldspar, which were suggested to exist but have never been directly obtained or measured at this resolution.

Thank you also for the suggestions in your comment #2. We have shortened the introduction and focussed more on the need for this study. We have also restructured section 2. However, we feel that the results from the multi-grain experiment (section 4.1) are an important argument in favour of the need for the present work, as it demonstrates the potential danger in measuring IR-RF on multi-grains. The vast majority of IR-RF studies have been on multi-grain aliquots, and the influence of the overlap of different signals from different grains has not received much attention in the literature so far. In section 4.1, we attempted to show through one example that the multi-grain measurements can be unreliable.

**Specific comments (authors' responses are shown in red after each numbered comment):**

On the role of Pb and Fe for feldspar luminescence (mostly section 2):

a.      Lines 91-92: I would suggest phrasing this more carefully, since Erfurt (2003) compared the emission spectra of a feldspar and KCl:Pb and inferred a relationship of the emissions due to the presence of Pb. Additionally, the two main oxidation states of Pb are $Pb^{2+}$ and $Pb^{4+}$ and a transition to $Pb^{+}$ via a highly reactive radical is rather unlikely. This could be a particular reason to wait for the XANES measurements for Pb oxides before further considering this hypothesis.

- Thank you for your comment. Here we just intended to state hypotheses based on previous work, but we have rephrased to avoid any misunderstanding: "The defect(s) responsible for the IR-RF emissions are still subject to debate. It has been suggested that the IR-RF emission occurs as a result of the change in the oxidation state of the participating defect via the transition: $Pb^{2+} \rightarrow (Pb^{+})^{*} \rightarrow Pb^{+}$ (Nagli and Dyachenko, 1986; Erfurt, 2003). A similar transition has been suggested for amazonite (see Ostrooumov, 2016), but the direct connection between the Pb-centre and IR-RF has not yet been evidenced. Other reactions involving $Pb^{4+}$ would also be possible but haven't yet been formally proposed.."

b.      Lines 97-98: Original references should be given here for the debate on Fe oxides in feldspars. A reference to include here could be Kirsh and Townsend (1988), especially because these authors also suggest further options for the role Fe might play for the red emission. The reference for the reaction of $Fe^{4+}$ to $Fe^{3+}$ due to the capture of an electron

would be Jain et al. (2015), although the presence of Fe4+ is rather unlikely in a natural mineral.

- Thank you, we have added the original references.

Line 111: Are these methods really the "best" suited methods to tackle the described problem? Either here or later in the paper, some advantages and disadvantages of using these methods should be highlighted. For instance, the penetration depth of the x-rays would be useful to know, especially, since in the conclusion a hypothesis of the effect of surface coating on luminescence is mentioned.

- Here, changed to: "μ-XRF and μ-XANES produce high-resolution maps of elements and their oxidation states and are well suited for the purposes of our study".

- For our samples, the 'penetration depth' is not entirely straightforward since we measured whole grains (unpolished), whose topography can change the depth from which the resulting XRF can still escape (attenuation length). Additionally, this depth depends on material density, which would be different for grains of different compositions as present in our dataset. Different elements will also have different attenuation lengths, e.g., for Fe, we consider 30–40 μm are probed. As the reviewer rightfully pointed out, this is a disadvantage of the method. For future work, we are considering using thin sections to avoid the differences in probing depths, surface effects and issues of self-absorption.

- In the main text, we added in section 3: "The XRF emission depth depends on the investigated element, the grain density and the angle of the stimulating beam. We selected relatively flat surfaces to avoid topography-artefacts. μ-XRF resulting from K in K-feldspar is expected to characterize the first ~10 μm of the grain, whereas the signal from Fe in K-feldspar characterizes the first 30–40 μm of the grain. The variable emission depth is an inherent disadvantage of our methodology."

Section 4.4: The paragraph is rather short and contains a contradiction. In line 237 it is suggested that Fe exists on the surface of the grain, but in line 238 an inclusion within the crystal is mentioned. Knowing the penetration depth of the x-rays and expanding this section, might help in clarifying this issue.

- We apologise for this misunderstanding. We wrote 'the surface of the grain' to highlight that our measurements are of the whole grain and we have no information about the composition of the total volume, but we are in fact, not measuring the surface, but a rim of the grain encompassing the top ~15% in depth (see comment above). We have rephrased: "These maps suggest that Fe exists in the top section of this feldspar grain in its Fe3+ and Fe2+ states. Note how Fe2+ is mainly clustered in one region, possibly within a mineral inclusion with a rim of Fe3+."

Technical comments (authors' responses are shown in red after each comment):

Line 26: A correlation is not statistically validated in this study; I would suggest exchanging the word "correlation" with "trend".

- Yes, 'trend' would be more appropriate in this case, thank you. Changed.

Line 76: It should read: "of twenty-one to…"

- Corrected.

Line 192-193: No references are given for the statement on the link of luminescence behaviour and volcanic origin of the samples. May I please ask the authors to delete this statement or to add a reference?

- We have added: "Common issues with volcanic samples are dim signals, different proportions of emissions, high fading rates and complex grain mineralogy (e.g., Krbetschek et al., 1997; Guérin and Visocekas, 2015; Joordens et al., 2015; Sontag-González et al., 2021; O'Gorman et al., 2021)."

Lines 199-202: Since the microXRF measurement results have not been described yet, I would suggest removing the statement of potential trends of K content and IR-Rf signal at this point.

- We have rephrased this paragraph to state that only a sub-population of grains displays the expected IR-RF signal and that our hypothesis is that only these are K-rich feldspar grains: "Further, our results demonstrate that a satisfying IR-RF signal can be measured for all our samples, but only by selecting grains with the appropriate luminescence characteristics (presumably K-feldspar grains). We hypothesize that the low-K grains with high Fe-content are the source of a contaminant IR-RF emission, which if not removed might result in a wrong equivalent dose estimation (i.e., a wrong age estimate)."

Lines 213-222: Maybe a boxplot or pie chart or some other type of illustration could be used to display the relative contributions of each element for each group of grains?

- Unfortunately, our methodology does not allow for the conversion of XRF intensity to elemental contribution (e.g., in weight) of the different elements. We can only plot relative XRF intensities of each element, as we used in Fig. 5, where we focussed on the five most relevant elements. We have added boxplots to Fig. 4 with the relative XRF intensities of each element for the three groups.

Fig. 5: Would it be possible to provide the reader with some numerical values for the axis of these ternary diagrams? Relative intensities are plotted, but on what basis? It would also be helpful, if the grains, which are displayed in Figure 6 could be somehow highlighted in Fig. 5.

- We have marked the grains from Fig. 6 in the ternary diagram as open symbols.

- The diagrams are plotted according to the XRF intensity (not the chemical composition), which only serves to compare grains to each other (e.g., grain A has more Fe than grain B), not to discern, e.g., which grains have more Fe than Pb in their compositions. We have added axes labels to the diagrams, but to avoid having the plots become too busy, we marked only the 50% point. A hypothetical point on the centre of the axis between, e.g., Fe and Pb would have the same XRF intensities (integrating counts in the deconvoluted peaks) for these two elements and no peak for the third element in the ternary diagram.

Fig. 7: Please indicate where this inset relates to the whole figure.

- The inset highlights the shift in the absorption edges for the three oxidation states. The units of the inset relate directly to the units of the main figure (the same underlying curves are shown, only the axis limits differ). This is now clarified in the caption: "The inset shows a magnification of the energy region relevant to determine the incident beam energies.."

Fig. 8: Are the areas displayed in Figure 8 the same as those in Figure 6? If yes, may I please ask the authors to highlight this in the figure caption. Alternatively, some overview images of the grains (similar to Fig. 6) could perhaps be given?

- It is the same grain in both figures, but the locations are slightly different. We now show the location using a blue square in Fig. 6.

Lines 250-252: Since no oxidation states of Pb were measured, it is unclear why this sentence is included in the conclusion of the paper.

- XANES measurements are possible for any element with XRF emissions within the measured energy limits. We highlighted the possibility of Pb oxidation states maps in the section 'Conclusions and future work' to suggest that they would also be informative to investigate the origin of the IR-RF, given the work by Erfurt et al., (2003). We have removed this suggestion from the sentence.

Line 254: It should be "element and oxidation state maps" and not mineralogical maps.

- Corrected.

Lines 258-260: Apologies if I missed it, but I think this is the first time that the hypothesis of elements on the surface of the grains influencing the luminescence signal is mentioned in the manuscript, except for the brief mentioning in section 4.4.

- We mention this possibility in section 4.1 and have expanded on this hypothesis: "We hypothesized that the unexpected signal increase originates from a different source, potentially from a coating around the grains due to the observation of a pinkish/reddish hue on some grains. Clay, Fe-oxide or other grain coatings are a common occurrence and additional preparation steps are sometimes undertaken to remove them prior to luminescence measurements (e.g., Jayangondaperumal et al., 2012; Lomax et al., 2007; Rasmussen et al., 2023). We attempted to remove this signal contamination, using different chemical treatments such as HF, regal water, and heated regal water, however, without success."

**Reviewer #2 Specific comments (authors' responses are shown in red after each numbered reviewer's comment):**

1.      Page 2 line 38, Introduction: "rely on the capacity of". Its not capacity, its property of defects. Change appropriately.

- Changed to 'property'.

2.      Page 2 line 40, 41, Introduction: "In the laboratory, the total amount of energy stored" and "energy absorption rate (dose rate, Gy a-1)". Energy per unit mass is dose.

- We have corrected both instances to 'energy per unit mass' and 'energy absorption rate per unit mass'.

3.      Page 2 line 43, Introduction: "quartz because of its high abundance and resistance to weathering". Besides these two, the fast to bleach OSL signal makes it most appropriate for geological dating.

- We have removed this sentence after the restructuring of sections 1 and 2, suggested by the other reviewer.

4.      Page 2 line 45, Introduction: "considering a low dose rate of 1 Gy ka-1". Why to mention 'low' here? Please delete low

- Done.

5.      Page 2 line 64, Introduction: "saturation cap at around 1500 Gy, reducing" cap can be deleted

- Done.

6.      Page 3 line 80, Method rationale: "study, between one and three grains (~…….. curves".  Statement not clear, consider revising. What is the reason for bad match? Bad match is often observed due to sensitivity changes. Pls refer (Varma, V., Biswas, R.H., Singhvi, A.K., 2013. Aspects of Infrared Radioluminescence dosimetry in K-feldspar. Geochronometria 40, 266-273.)

- Changed 'between one and three grains' to 'between one and three grains per sample' to clarify and added the suggested reference to explain the bad match.

7.      Page 3 line 83, Method rationale: "grains that emitted a detectable signal displayed the expected decay shape." What is meaning of expected shape here? Exponential decay?

- Changed 'expected decay shape' to 'decay shape expected for IR-RF (decreasing signal with increasing dose)'

8.      Page 3 line 89, Method rationale: "With dose exposure, the 955 nm emission increases and overlaps with the 880 nm peak." Why is it so? How dose increases the 955 nm emission, does it mean more 955nm recombination centres are being regenerated? Does it reflect multiple trap system?

- There is a mistake in this sentence; it is the ~700 nm emission that increases with dose, not the 955 nm. We apologise for this oversight, we replaced the sentence with "… which partly overlaps with the 880 nm peak".

9.      Page 4 line 101, Method rationale: "The thermal stability of the ~710 nm emission has been, however…. the measured IR-RF" The red TL emission in feldspar is generally considered more stable than conventional IRSL method, why the red IR-RF is unstable?

- To the best of our knowledge, no study can be found in the literature specifically investigating the differences of the red-TL and red-RF emissions. Prasad and Jain (2018) described that there are at least two and possibly more components in the red photoluminescence emission, so it's possible that different signals are obtained with TL and RF. This seems to be in line with the findings from Dütsch and Krbetschek (1997) and Krbetschek et al. (2002) who found that the red-RF peak positions shift

with potassium concentration. Another considerable experimental difference is that the red RF is obtained at a relatively low temperature of (up to) 70°C (if following the current dating protocol), whereas the red TL peak is above ~200°C (e.g., Zink and Visocekas, 1997).

Prasad, A.K., Jain, M. Dynamics of the deep red Fe3+ photoluminescence emission in feldspar. Journal of Luminescence 196, 462–469. https://doi.org/10.1016/j.jlumin.2017.11.051, 2018.

Dütsch, C. and Krbetschek, M. R.: New methods for a better internal 40K dose rate determination, Radiation Measurements, 27, 377–381, https://doi.org/10.1016/s1350-4487(96)00153-9, 1997.

Krbetschek, M. R., Götze, J., Irmer, G., Rieser, U., and Trautmann, T.: The red luminescence emission of feldspar and its wavelength dependence on K, Na, Ca – composition, Mineral. Petrol., 76, 167–177, https://doi.org/10.1007/s007100200039, 2002.

Zink, A.J.C., Visocekas, R. Datability of sanidine feldspars using the near-infrared TL emission. Radiation Measurements 27, 251–261. https://doi.org/10.1016/S1350-4487(96)00141-2, 1997.

10.    Page 4 line 111, Method rationale: "The μ‑XRF and μ‑XANES techniques are best suited for this purpose by producing high-resolution maps of elements and their oxidation states". It will be good to provide some details about the mentioned techniques and their usefulness for present work.

- We added the following to introduce the two techniques: "μ‑XRF elemental analyses are based on the characteristic fluorescence of atoms when stimulated with X-rays with a higher energy than their ionisation energy. In the case of μ‑XANES, initial measurements of standards are run by varying the incident beam energy to determine the specific energy equal to the absorption edge (binding energy of inner shell electrons) of the element or ion of interest. This is apparent by an abrupt rise in the resulting fluorescence, which is different between oxidation states as they require different minimum stimulation energies before ionisation and subsequent fluorescence. μ‑XRF maps using the obtained absorption edge energies allow for maps of the different oxidation states of the same element."

11.    Page 3 line 83, Method rationale: "The use of synchrotron μXRF allows us to improve the spatial resolution compared with previous uses of μXRF (e.g., Buylaert et al., 2018) by reducing the beam spot size from ~25 μm to 1 or 0.5 μm". It is indeed impressive that spot size is smaller and we can work at higher resolutions, but how will it effect S/N ratio and thus elemental concentration estimation? In addition, since spot size is smaller, only few grains analysis may be possible. In such cases, how can we get the statistical representation of entire grain population just based on few grains studies?

- The signal to noise ratio can be optimised by changing the dwell time (signal integration time) for each position without loss of spatial resolution. We did not have issues with the signal to noise ratio for the μXRF spectra. However, as the reviewer pointed out, the length of measurements at high spatial resolution is indeed a hindrance to obtain a statistically significant representation of a sample. For this study, we chose a high spatial resolution on a small number of grains to investigate what effects small-scale heterogeneities might have, but this approach might not be appropriate for other studies.

12.     Page 4 line 128, Material and instrumentation: Normally Tuff samples are expected to contain Fe rich species. Is this a deliberate choice to look the effect of Fe in the samples as 2 out of 5 are tuff samples?

- It wasn't a deliberate choice to investigate Fe-rich samples. The aberrant IR-RF curves were observed by chance when dating samples from Kenya, particularly from volcanic regions that are often challenging for dating. We chose these samples to investigate the reasons behind the unexpected IR-RF curves and because of the volcanic aspect, Fe was an obvious choice of element for us to investigate further. Other samples (e.g., Gi326, H22550) were chosen as 'reference' samples for comparison.

13.     Page 5 line 134, Material and instrumentation: Why only the sample H22550 was etched with HF. Why not same is performed for other samples?

- Sample H22550 was prepared by a different laboratory, which includes HF etching in their pre-treatment of K-feldspar. We chose to include the sample despite the slightly different pre-treatment because this sample was used in an important study that assessed the accuracy of IR-RF dating (Buylaert et al., 2012), and, thus, served as a good comparison sample.

14.     Page 5 line 134, Material and instrumentation: "Multi-grain and single-grains … National Laboratory" How correspondence between IRRF and XRF signals is established?

- Individual grains were first measured on a *lexsyg research* luminescence reader to obtain IR-RF (one grain at the centre of a cup) and then transferred to sticky carbon tape on a polymer slide for µXRF measurements. The position of the grains of the slide was carefully written down to correspond both signals for each grain. No µ-XRF was performed on the multi-grain aliquots. We rephrased: "After IR-RF measurements, the grains were removed from the stainless steel cups and fixed on a polymer microscope slide (UVT acrylic; Agar Scientific) with a small piece of carbon tape to avoid misplacement during measurement (supplementary Fig. S1)."

15.     Page 5 line 143, Material and instrumentation: "2016). Grains were fixed on a polymer microscope slide…………" What are spectral and luminescence characteristics of the base material used?

- For the luminescence measurements, we used the standard Freiberg Instruments stainless steel sample cups; the background level (cup emission + system) is shown in Fig. 2e. The XRF emission depth is dependent on grain density and element, but the signal is expected to be dominated by the top-most volume. For Fe, the XRF signal is expected to characterise only the first 30–40 µm of the grains, which is relatively small compared to the grain diameter of at least 90 µm, so we don't expect any contribution from the base material. We added the information on the emission depth, as detailed in the specific comments of reviewer #1.

16.     Page 5 line 144, Material and instrumentation:  "XRF maps were obtained by scanning across pre-selected regions on the grains 90 x 90 µm maps,"  What is the basis of ROIs selection?

- We selected square regions on the grains with low topographic changes (i.e., as flat as possible). Added to text.

17.     Page 5 line 149, Material and instrumentation: "resolution of 0.67 μm was achieved by focusing the beam with" Is it Xray beam focussing or luminescence focussing, please specify.

- Only the X-ray beam is focussed onto the sample position to achieve the desired spot size. Changed to: "μ‑XRF maps were obtained by scanning across pre-selected regions on the grains with low topographic changes (90 x 90 μm maps, with a step size of 0.67 μm and an integration time of 0.1 s). The incident X-ray beam was focussed by a pair of Kirkpatrick-Baez mirrors."

18.     Page 5 line 150, Material and instrumentation: " An incident beam energy of 13.5 keV was" Why this energy chosen any specific reason?

- The incident beam energy determines which elements are ionized, i.e., the beam energy must be higher than the ionisation energies of the elements of interest. There are also elastic and inelastic scatter background peaks that appear near the incident beam energy in the XRF spectrum, so we chose 13.5 keV to provide some separation between the background peaks and the Pb peak that appears at 10–11 keV.

19.     Page 5 line 150, Material and instrumentation: "fluorescence was detected through the sum of 4 silicon drift detectors" Why these four detectors were used? Can we use PMT instead? whats the advantage we get with use of these detectors. Can you provide geometry of measurements and experimental setup?

- These detectors form part of the beam line setup and cannot be replaced for XRF measurements. With the silicon drift detector, we are able to obtain the energy of incoming photons, needed for subsequent analyses. In contrast, PMTs are generally capable of detecting low-level light, so can not be used for detecting X-rays (multi-keV-range). We rephrased the sentence in question: "The excited elements' characteristic fluorescence was obtained from the sum of the four elements of a silicon drift detector."

- The experimental setup is shown in the image below and more details on the synchrotron beam line can be found in Chen-Wiegart et al. (2016).

[Figure]

(Image: https://www.bnl.gov/nsls2/beamlines/beamline.php?r=5-ID)

Chen-Wiegart, Y.K., Williams, G., Zhao, C., Jiang, H., Li, L., Demkowicz, M., Seita, M., Short, M., Ferry, S., Wada, T., Kato, H., Chou, K.W., Petrash, S., Catalano, J., Yao, Y., Murphy, A., Zumbulyadis, N., Centeno, S.A., Dybowski, C., Thieme, J., 2016. Early science commissioning results of the sub-micron resolution X-ray spectroscopy beamline (SRX) in the field of materials science and engineering. AIP Conference Proceedings 1764, 030004. https://doi.org/10.1063/1.4961138

20.    Page 5 line 152, Material and instrumentation: "All XRF measurements were normalised to the corresponding incident X-ray flux " X-ray sources are normally found inhomogenous spatially and temporally. Does this can effect your measurements?

- Low-emittance synchrotron radiation as delivered by NSLS-II allows beam lines to focus X-rays to an almost diffraction-limited spot, thus avoiding any potential spatial inhomogeneities. Third-generation synchrotron radiation facilities deliver X-rays in an extremely stable beam. Nonetheless, the incident beam intensity is continuously monitored and used to correct for minute X-ray flux changes (example shown in Fig. S2). By comparing Fig. S2a and c (before and after correction for flux fluctuation, respectively), we note that the effect is almost imperceptible in our maps.

21.    Page 5 line 150, Material and instrumentation: "The XANES maps had a resolution of 0.5 μm (60 x 60μm)." Are units correct? How does 60 um X 60 um translate to 0.5um? not clear.

- The units are correct. We have rephrased: "The μ-XANES maps cover an area of 60 x 60 μm in steps of 0.5 μm, thus creating a grid with 120 x 120 data points (i.e., 14 400 spectra)."

22.    Page 5 line 150, Material and instrumentation: "we varied the incident beam energy according to the absorption edge values obtained from the μXANES measurements of Fe standards (Fe foil, pyrite, hematite)." How the specific absorption edge values were estimated?

- The spectra from the μ-XANES measurements of Fe foil, pyrite and hematite were processed using the software ATHENA v.0.9.26 (Ravel and Newville, 2005) and are shown in Fig. 7. The portion of the spectra containing the absorption edges is highlighted in the inset of Fig. 7. The specific absorption edges (i.e., the excitation energy necessary to eject the electron) were directly read from that figure and are marked with dashed lines for $Fe^{2+}$ and $Fe^{3+}$. Since the absorption edge of $Fe^{3+}$ is higher than for $Fe^{2+}$, an XRF measurement at that energy will contain emissions from both $Fe^{3+}$ and $Fe^{2+}$.

23.    Page 5 line 150, Material and instrumentation: "(i) the total Fe (at 7.275 keV)," The energies mentioned here are quite precise. How much is normally the resolution. Since electronic energy levels of specific elements are quite low in energy (~few eVs) compared to what is being provided, then why this much precision is needed?

- The energy resolution around the Fe K-absorption edge is below 1 eV. The energy difference between the $Fe^{2+}$ and $Fe^{3+}$ absorption edges is of 12 eV (equal to 0.012 keV) (see Fig. 7 and line 177 of the revised manuscript), so this precision is needed to differentiate between the two oxidation states.

24.      Page 6 line 165, Results: "500 Gy succeeded by an increase, roughly following a saturating exponential shape" What is reason behind increase to saturating exponential behaviour? Why should there be an increase at all considering physics aspect? What is the nature of sample X7343, Is it similar to volcanic tuff?

- Sample X7343 is from a volcanic region, but not directly from a volcanic deposit. Our study could not conclusively define the reason behind the increasing RF behaviour, but we determined that the dual (decreasing then increasing) behaviour of the multi-grain aliquots was caused by superposition of signal from different grains, which in turn were either only increasing or only decreasing. We observed that grains with relatively high Ca and high Fe content tended to display the increasing RF behaviour. It is possible that the increasing signal observed in the IR region is in fact only the tail of the far-red emission associated with Fe, which is known to have this behaviour (see section 2).

25.      Page 6 line 169, Results: "contamination, potentially coming from coating around the grains, we" Why is it assumed that coating could be responsible?

- We observed that the grains had a reddish hue, so hypothesised that an iron coating could be affecting the signal.

26.      Page 6 line 171, Results: "Despite using density … high Fe content". Does that mean it is Na or Ca feldspar grains? Have you performed XRD analysis on bulk to see the mineralogy of samples?

- We have not performed any other analyses on these samples, so cannot say what the bulk composition is. However, our work indicates that a bulk-approach might be insufficient to describe these samples, given their heterogeneous luminescence behaviour at single-grain level.

27.      Page 6 line 176, Results: "their visual appearance under a microscope" What were the visual indicators considered for choosing K-Feldspar?

- The visual indicators are given in line 196-197 of the revised manuscript: "Between 10 to 30 grains were placed onto two aliquots, one for transparent shiny angular grains and one for white-pinkish rounded grains." As shown in Fig. 1, the transparent shiny angular grains gave a signal corresponding to that expected from K-rich feldspar.

28.      Page 6 line 178, Results: "grains. The regenerated IR-RF signals showed a clear distinction between the two aliquots (Fig. 1), proving it is possible to separate the two observed IR-RF shapes." This is quite a qualitative way. I am not sure how to progress using only visually inspected grains. The visual appearance and selection can vary depending upon geological settings of grains as well as person observing them. Is there any other rigorous way of making such selection?

- The visual appearance only served as proof of concept and we do not recommend it for future applications. As stated in the conclusions (line 269-271 in the revised manuscript), "manually picking individual grains was necessary to isolate K-feldspar grains, which is unrealistic for routine dating applications in a low-light laboratory. A more realistic way to remove such contamination is by selecting K-rich feldspar grain

populations by isolating the emission signal of individual grains with an imaging system."

29.    Page 6 line 184, Results: "presumed to be the low-K, Fe rich minerals identified via SEM-EDS" Low K means possibly high Na or Ca, why only Fe is considered. Fe if present should be in form of defects, which should be in ppm level. Can uXRF measure to such low concentration levels? If Fe is appearing as major element in feldspar separates, it means it is present in stoichiometric formula and in that case, mineralogy of sample would be different. Please suggest if I am missing something.

- The SEM-EDS Fe-content of sample X343 ranged 0.4–22.4 wt.%. Unfortunately, our μXRF measurements are not calibrated to provide weight percentages, but, in general, XRF measurements can detect elements down to ppm-concentrations. Both the SEM-EDS and μXRF analyses were of whole grains, so it is possible that the Fe-rich grains are "normal" feldspar grains (i.e., with no stoichiometric Fe) but with an Fe coating contributing to the signal. It is also possible that these are grains of an entirely different Fe-rich mineral. Further work is required to identify the mineral(s) with aberrant IR-RF behaviour.

30.    Page 6 line 186, SG IR-RF characterisation: "signal of twenty-two individual grains coming" Can you specify mineralogy of each grains, which are picked for such measurements?

- These were "new", previously unmeasured grains, so we do not have SEM-EDS analyses for them.

31.    Page 6 line 165, SG IR-RF characterisation: "2): Category #1 for grains with a decreasing IR-RF signal, category #2 for grains with an increasing IR190 RF signal, and category #3 for grains with a flat signal". How many grains falls in each category and is there any link to the provenance.

- The numbers of grains are listed in Table 1 (9, 8 and 5 grains, respectively, in each category). The Kenyan samples had a higher proportion of category #2 grains, but the numbers are too small to make any meaningful interpretations of provenance.

32.    Page 6 line 165, SG IR-RF characterisation: "we also observed the unwanted decreasing IR-RF signal for one of the four grains for sample H22550, which is from a coastal sand deposit." We expect IR-RF signal to decrease with irradiation, so why it is said unwanted ?

- We apologise, we meant "the unwanted increasing…"

33.    Page 6 line 193, SG IR-RF characterisation: "When the total signal of the theoretical aliquot was composed of more than 50% of signal from the category #2 grain, we observed the same decay shape as in figure 1 for a multi-grain aliquot sample X7343" Obviously, since the two different category of grains having two different IR-RF characteristics are being added, so the result will depend on the proportion of the individual populations in the mixture. More importantly, it is important to know, how these two grains are different with respect to crystallography or stoichiometry or defects concentration. Is the nature of curve repeatable over repeated bleaching and irradiation cycles?

- Future work should indeed investigate in detail the mineralogy, crystallography and trace elements in the grains leading to the two signals. However, we do not agree that the mixture of decay shapes is trivial. We modelled that the IR-RF signal from a multi-grain aliquot with up to 20% contaminant grains would not be flagged as abnormal (i.e., it would still display only a decreasing signal), but the IR-RF characteristics relevant for dating (e.g., saturation, stability) would already be impacted. This finding might explain the mixed accuracy observed for IR-RF ages. The curve shape is repeatable.

34.     Page 7 line 201, SG IR-RF characterisation: As mentioned by authors, long-term signal stability may not be there for bad traps, is there a way to prove it? How do we know it without experiment?

- We have not yet conducted IR-RF fading tests for these samples, but merely mentioned what the consequences would be of different luminescence behaviours between the two groups.

35.     Page 7 line 204, SG IR-RF characterisation: "Further, our results demonstrate…" I agree with this statement, but it is still not clear how can we segregate K-Feldspar and other minerals. Manually it will not be possible on routine basis.

- Using an imaging system, as suggested in the conclusion (lines 270-272 in the revised manuscript), would solve this issue. Dose-response curves can be obtained for all grains and only those with appropriate characteristics would be used for $D_e$ estimation.

36.     Page 7 line 207, Subgrain µXRF elemental maps: "We then fitted each of the **18 225** spectra for…." this statement is not clear

- We meant eighteen thousand two hundred and twenty five spectra: one spectrum for each pixel in a grid of 135 by 135 pixels with step sizes of 0.67 µm, covering an area of 90 by 90 µm. Each of these spectra was fitted to deconvolute the emission peaks of different elements. Changed to : "We then fitted each of the spectra in the 135 by 135 pixel grid (i.e., 18 225 spectra)…"

37.     Page 7 line 212, Subgrain µXRF elemental maps: "characterise visible inclusions (see Table 1)." The number of grains analysed per samples are quite small to represent the statistics of system. Can we consider them as representative of whole samples? It is difficult to conclude unless sufficient data points exists.

- It is true that we had only a small number of grains per sample. However, joining all samples and categorizing by IR-RF behaviour gave us 9 and 8 grains in groups #1 and #2, respectively, which we believe to be sufficient for this pilot study. We observed the trend that grains with the desired IR-RF signal shape had high Pb and Ba and low Fe and Ca contents. In future work, we will increase the number of grains.

38.     Page 7 line 214, Subgrain µXRF elemental maps: "all contain K, Pb, Fe and Ba, among other elements (Fig. 4)." What is typical concentration of these elements? Considering K is a major element present in stoichiometric formula, how much is relative concentration of the other elements?

- Unfortunately, our methodology does not allow for the conversion of XRF intensity to elemental composition (e.g., in weight) of the different elements. We can only compare the XRF intensities of each element, between different grains, as done in Fig. 5.

39.     Page 7 line 223, Subgrain µXRF elemental maps: "grains from category #3 cluster relatively close to those from category #2, suggesting" How and why does this effect the IR-RF properties? These are observations, but what is the reason for IR-RF signal due to such clusters is not clear.

- We do not have a definitive explanation for the different IR-RF behaviours in grains of similar elemental compositions. We suggested that "elemental composition alone is not responsible for the lack of IR-RF signal but that the spatial configuration of the elements is another important factor to consider" (line 247-248 in the revised manuscript). By this, we mean that the presence of the same elements at different sites in the crystal might elicit different luminescence responses. Site-selective luminescence behaviour has been reported previously (e.g., Kumar et al., 2020).

40.     Page 7 line 237, Mapping oxidation states with µXANES: "suggest that Fe exists on the surface of this feldspar grain in its Fe3+ and Fe2+ states." It is great that using uXANES, we could map the presence of Fe on the surface of feldspar grain, but luminescence or IR-RF is a volumetric phenomenon. How this observation is helpful in explaining the IR-RF signal.

- In future work, we aim to set up a dual detector to detect simultaneously XRF and the IR-RF resulting from the X-ray irradiation. Such a setup would enable a direct comparison.

**Additional changes by authors:**

1) Updated author affiliation.

2) Corrected number of grains measured by SEM-EDS (118) and changed colours in Fig. 1.

3) Changed the microscope images to sharper SEM images in Fig. 6.

4) Updated Fig. 7 with less noisy spectra of the same samples.

5) Added acknowledgement of NSLS-II facilities.

6) Minor text corrections throughout the main text.

---

## Referee Report (RR1)

**Report on "Short communication: Synchrotron-based elemental mapping of 1 single grains to investigate variable infrared-radiofluorescence 2 emissions for luminescence dating"**

It is good to see that authors have significantly improvised the earlier version of manuscript. They have revised and clarified most of the scientific inconsistencies. The manuscript now reads well, measurement, results, and discussions are linked properly. There are several important aspects of manuscript, which will be useful for luminescence studies in future. The authors have done hard work in conducting measurements of individual grains and conducting measurements at sub-microscopic level for each grain. The finding related to correlation of elemental concentrations with anomalous IR-RF signal is interesting and needs further explorations. Thus, I feel manuscript should be published.

Although the major findings are interesting, yet I feel that manuscript still needs improvement in the English and clarity on scientific statements. I would like to encourage authors to read and write carefully to remove the inconsistencies. My comments are given below and points out some of the mentioned contradictions.

**Comments**

1. Line 43 (dose (Gy))
2. Line 44 sampled sediments
3. Line 71, predicted instead of assumed
4. Line 76, It will be good to state some reasons which could be leading to the variabilities rather than just mention a term.
5. Line 78, "what…" either put ? in end or rephrase
6. Line 86, "identification….." Is it right to presume and specify signal as contamination without justifying it. You may say that u-XRF and u-Xanes have capabilities to identify defects and isolate signals but presuming that it's a contaminating signal is not right without proving.
7. Line 97 "Though both …..geochemistry. " Authors pointed out important aspect in this statement that both crystallography as well as geochemistry is playing a role here but they are only focusing of geochemistry. So how it is justified to attribute the current study observations only to geochemistry without considering crystallography?
8. Line 103 "Other reactions…….". why do you say so??? May be it does not participate thatswhy not observed. I think it's better to rephrase as "involving higher oxidation states might be playing a role but not observed or suggested"
9. Line 122 "Such a contribution….De. " In this case relative intensities play a vital role. If authors say it is only 5%, then it shouldn't significantly effect estimated dose. Please see numbers very carefully.
10. Line 144 "between one and three grain." Is 'and' right here?
11. Line 174-176 "We paid particular attention…….." Do you mean other regions are not responding to IRRF? what about other regions? How are you sure that these regions are responsible for anomalies.
    What about IR RF characteristics of Na? Is it not important?
    Na/K Feldspar normally are dominant remnants in feldspar separation process.
12. Line 231-232 "We hypothesized……". How good is this hypothesis. Is it not creating biasing? Normally Fe coating on the grains can be dissolved by long HCl treatment as suggested by

Jayangondaperumal et al., 2012. Does that not mean that it may not the external Fe coating? Sometimes K-Feldspar also have a pinkish hue and Na-Feldspar is white. Hope this is not creating issue for authors as Na is not being measured in current methods.

13. Line 240-241, "Note that…." I raised this concern in my earlier comments also and didn't get a satisfactory answer for this. Please refer comment 29. If Fe is in coating, it can be dissolved by HCl/HF treatment. If it is in the volume and goes as high as 10% then it should be part of stoichiometry formula as is case with K. This can change the interpretations. Authors must see to this aspect.

14. Line 240-241 "Between 10 to 30………….." I know it's difficult, but it is really important to establish the mineralogy of these grains. As authors used transparent shiny and pinkish white grains, which could be Na Feldspar and K-Feldspar respectively. This could really effect the interpretations.

15. Line 246-250: How does the manual picking and optical microscope observations related with respect to i) , ii) and iii).

16. Line 273-276 "Further….." It is difficult to accept that it is pure K-Feldspar. how does correlation stands for K%? Please verify on some standard sample.
    This is a hypothesis, and it is difficult to justify unless it is quantified. At present, we can only relate it to the color, which authors are expecting due to Fe content. The authors must understand even K-Feldspar has pinkish appearance and Na-Feldspar has whitish appearance. Although other colours does exist. Therefore, it may not be right to attribute the colour to Fe content without quantification.

17. Line 284-285 "grains displaying…." This statment indicate that elemental concentration K, Pb, Fe and Ba are not representative IR RF signal and should not be correlated to signal shape. On the other hand Ca, Ti, Mn may be related.

---

## Author Response (AR2)

We thank the reviewers and the associate editor for their feedback. Below, we address each comment individually.

**Reviewer #2 (authors' responses are shown in red after each numbered comment):**

It is good to see that authors have significantly improvised the earlier version of manuscript. They have revised and clarified most of the scientific inconsistencies. The manuscript now reads well, measurement, results, and discussions are linked properly. There are several important aspects of manuscript, which will be useful for luminescence studies in future. The authors have done hard work in conducting measurements of individual grains and conducting measurements at sub-microscopic level for each grain. The finding related to correlation of elemental concentrations with anomalous IR-RF signal is interesting and needs further explorations. Thus, I feel manuscript should be published.

Although the major findings are interesting, yet I feel that manuscript still needs improvement in the English and clarity on scientific statements. I would like to encourage authors to read and write carefully to remove the inconsistencies. My comments are given below and points out some of the mentioned contradictions.

Comments

1. Line 43 (dose (Gy))

   ● Changed to "(dose, with the unit Gy)."

2. Line 44 sampled sediments

   ● Change incorporated.

3. Line 71, predicted instead of assumed

   ● Change incorporated.

4. Line 76, It will be good to state some reasons which could be leading to the variabilities rather than just mention a term.

   ● Added: "(e.g., differences in signal saturation or in proportions of RF emissions)"

5. Line 78, "what…" either put ? in end or rephrase

   ● Rephrased to: "We […] discuss the effect that the observed variability could have on multi-grain aliquots."

6. Line 86, "identification….." Is it right to presume and specify signal as contamination without justifying it. You may say that u-XRF and u-Xanes have capabilities to identify defects and isolate signals but presuming that it's a contaminating signal is not right without proving.

   ● Indeed, at this point in the text we have not yet introduced the potential IR-RF contamination, so we have rephrased the sentence (changes in bold type): "… identification of the origin of **RF emissions** could help us to gain a better understanding about apparent saturation or quenching of the IR-RF signal."

7. Line 97 "Though both …..geochemistry. " Authors pointed out important aspect in this statement that both crystallography as well as geochemistry is playing a role here but they are only focusing of geochemistry. So how it is justified to attribute the current study observations only to geochemistry without considering crystallography?

- Unfortunately, it was beyond the scope of this work to include crystallography, but we have rephrased the sentence to highlight that this should be done in the future and compared against our geochemical results: "Both the grain geochemistry and crystallography should be investigated to characterize the defect type and its environment. In the present study, we focussed only on geochemistry, though our results should be complemented with crystallographic studies in future work."

8. Line 103 "Other reactions…….". why do you say so??? May be it does not participate thatswhy not observed. I think it's better to rephrase as "involving higher oxidation states might be playing a role but not observed or suggested"

- We have incorporated the reviewer's suggestion: "Other reactions involving higher oxidation states would also be possible but have not yet been observed or formally proposed."

9. Line 122 "Such a contribution….De. " In this case relative intensities play a vital role. If authors say it is only 5%, then it shouldn't significantly effect estimated dose. Please see numbers very carefully.

- Due to the flattening of the IR-RF curve as it nears saturation, small changes in signal can have a high impact on $D_e$. This statement is based on observations in Sontag-González and Fuchs (2022), cited in that sentence, where $D_e$ changes of ~400 Gy were caused by signal changes of only a few percent. We have added in the main text that the effect is on $D_e$ values "at doses near signal saturation" to clarify this issue.

10. Line 144 "between one and three grain." Is 'and' right here?

- Yes, we mean 1–3 grains per sample.

11. Line 174-176 "We paid particular attention…….." Do you mean other regions are not responding to IRRF? what about other regions? How are you sure that these regions are responsible for anomalies.

What about IR RF characteristics of Na? Is it not important?

Na/K Feldspar normally are dominant remnants in feldspar separation process.

- Ideally, we would have preferred to use an instrument that also measured Na, since it is an important element in the feldspar ternary series, as mentioned by the reviewer. However, the SRX beamline is not set up for XRF analyses of elements with low atomic numbers. In the literature, Na has not been linked to the infrared but red/green RF emissions. Therefore, we focussed on Fe and Pb. We have no spatially-resolved IR-RF information, so we do not know in which regions in the grain the RF emissions originate. We are also not sure from which element or mineral the contaminant emission observed in the IR-RF is originating. A motivation of the current manuscript was to test whether there was a trend between the IR-RF curve shape and either Fe or Pb content, among other elements. Indeed, our µ-XRF results support a link between Fe content and the unwanted IR-RF curve shape (category #2 grains). For more conclusive or definitive results, more work is needed on a larger number of grains and a

combination of different techniques would be advisable to characterise all elements of interest.

- For one sample (X7343) we have SEM-EDS measurements (introduced in section 4.1; data is available open-access). This technique allows for the quantification of Na. As shown below, for this one sample we see that the Fe-rich grains (plotted in green and blue) are present in grains low in Na and in K (all grains were low in Ca, i.e., <1wt%). Though the Fe-rich grains characterised by μ-XRF that showed a trend with the unwanted IR-RF were in a different set of grains, the combination of both datasets suggests the unwanted grains are not Na-rich.

[Figure]

**X7343 SEM-EDS**

12. Line 231-232 "We hypothesized……". How good is this hypothesis. Is it not creating biasing? Normally Fe coating on the grains can be dissolved by long HCl treatment as suggested by Jayangondaperumal et al., 2012. Does that not mean that it may not the external Fe coating? Sometimes K-Feldspar also have a pinkish hue and Na-Feldspar is white. Hope this is not creating issue for authors as Na is not being measured in current methods.

- We tested this hypothesis using similar treatments to that suggested by the reviewer and observed no difference in IR-RF curve shape. This result could be due to insufficient treatment, so we cannot entirely rule out the possibility of the signal emanating from a coating. However, we have added a conclusion to our hypothesis based on our results (changes in bold type): "We attempted to remove this signal contamination using different chemical treatments such as HF, regal water, and heated regal water, however, without success. **This suggests the signal is not originating from a coating.**"

13. Line 240-241, "Note that…." I raised this concern in my earlier comments also and didn't get a satisfactory answer for this. Please refer comment 29. If Fe is in coating, it can be dissolved by HCl/HF treatment. If it is in the volume and goes as high as 10% then it should be part of stoichiometry formula as is case with K. This can change the interpretations. Authors must see to this aspect.

- The minerals with up to 22wt% Fe (determined by SEM-EDS) probably have Fe in their stoichiometric formula, but these are probably not feldspar minerals because the Fe content is too high (see also plot in comment 11). These might be inclusions of other minerals in a mostly K-rich feldspar matrix or these might be Fe-rich minerals grains with small inclusions of feldspar and/or other minerals. The SEM-EDS measurements are only of spots (not maps) so we cannot conclusively distinguish these two scenarios. We note that mineralogically heterogeneous grains have been described for volcanic sediments (e.g., O'Gorman et al., 2021), which make density separation especially unreliable to isolate K-rich feldspar grains.

- From our μ-XRF maps (Fig. 6), we can see that none of the three elements (K, Fe and Pb) are homogeneously distributed. In the category #1 grain (top row), Fe appears in higher concentrations in areas with low K concentrations (this is apparent from the lack of dark blue or black areas). In the category #2 grain (middle row), most of the grain contains only Fe and no K or Pb (magenta); in the one region containing K, only a small area contains both K and Fe (dark blue). These results support our conclusions that Fe is not a stoichiometric element in feldspar grains but appears in more or less discrete locations.

- In reference to comment 29 of the previous round, XRF measurements would theoretically be able to detect concentrations down to a ppm range. However, in order to obtain true concentrations, the sample would need to be thinner than 1 μm to avoid depth-effects and self-absorption, so we cannot make a statement as to the concentration in our grains. Additionally, as we have shown in comment 11 (and the plot therein), a low-K grain does not necessarily mean high Na or Ca.

14. Line 240-241 "Between 10 to 30.............." I know it's difficult, but it is really important to establish the mineralogy of these grains. As authors used transparent shiny and pinkish white grains, which could be Na Feldspar and K-Feldspar respectively. This could really effect the interpretations.

- As highlighted in the conclusions (section 5), we do not recommend manually picking grains based on visual characteristics. We agree that establishing the mineralogy is important, which is why we introduced the μ-XRF analyses in section 4.3. Future work should indeed use complementary techniques (e.g., SEM-EDS) to also characterise the Na-content of grains, even though the spatial resolution would not be comparable to that from the μ-XRF.

15. Line 246-250: How does the manual picking and optical microscope observations related with respect to i) , ii) and iii).

- We have rephrased to clarify that, after manual picking, we measured the two subsets of grains, i.e., a smaller subset that yielded the expected decreasing IR-RF curve and a larger subset, which yielded an increasing IR-RF signal (changes in bold type): "By manually selecting the grains based on their shape and colour a**nd the subsequent multi-grain IR-RF measurements**, we made three important observations:…"

16. Line 273-276 "Further….." It is difficult to accept that it is pure K-Feldspar. how does correlation stands for K%? Please verify on some standard sample.

This is a hypothesis, and it is difficult to justify unless it is quantified. At present, we can only relate it to the color, which authors are expecting due to Fe content. The authors must understand even K-Feldspar has pinkish appearance and Na-Feldspar has whitish appearance. Although other colours does exist. Therefore, it may not be right to attribute the colour to Fe content without quantification.

- The attribution to K-feldspar is based on the IR-RF curve shape of these grains and not on the grain colour. All IR-RF literature reports IR-RF and *decreasing* IR-RF signals for K-feldspar and this curve shape was only observed for one of the two groups of grains, which happened to be transparent shiny angular grains. The other group displayed a different IR-RF curve shape (i.e., increasing), which has yet to be reported for K-feldspar grains. Since many previous IR-RF studies of K-feldspar samples with known compositions are available (e.g., Trautmann et al., 1998; Kumar et al., 2018; Murari et al., 2021), we do not find it necessary to measure standards to confirm that K-feldspar displays a decreasing curve shape as it is the fundamental assumption of IR-RF measurements.

17. Line 284-285 "grains displaying…." This statment indicate that elemental concentration K, Pb, Fe and Ba are not representative IR RF signal and should not be correlated to signal shape. On the other hand Ca, Ti, Mn may be related.

- In this sentence, we are only considering the presence/absence of elements. It is true that the presence of K, Pb, and Fe cannot be correlated to signal shape, since most grains contain them. We have changed the text to highlight this: "Most grains across all categories contain K, Pb, and Fe, and other elements. Among the grains displaying a decreasing IR-RF signal (category #1), all contain Ba (Fig. 4, middle), which is less present in grains from categories #2 and #3." However, we note that later in the text we show that the relative proportions of the common elements appears to be correlated to signal shape. For example, in Fig. 5a, grains from categories #1 and #2 cluster separately (with a few exceptions) even though only K, Pb, and Fe are considered.

**References**

Kumar, R., Kook, M., Murray, A. S., and Jain, M.: Towards direct measurement of electrons in metastable states in K-feldspar: Do infrared-photoluminescence and radioluminescence probe the same trap?, Radiation Measurements, 120, 7–13, https://doi.org/10.1016/j.radmeas.2018.06.018, 2018.

Murari, M. K., Kreutzer, S., Frouin, M., Friedrich, J., Lauer, T., Klasen, N., Schmidt, C., Tsukamoto, S., Richter, D., Mercier, N., and Fuchs, M.: Infrared Radiofluorescence (IR-RF) of K-Feldspar: An Interlaboratory Comparison, Geochronometria, 48, 105–120, https://doi.org/10.2478/geochr-2021-0007, 2021.

O'Gorman, K., Tanner, D., Sontag-González, M., Li, B., Brink, F., Jones, B. G., Dosseto, A., Jatmiko, Roberts, R. G., and Jacobs, Z.: Composite grains from volcanic terranes: Internal dose rates of supposed 'potassium-rich' feldspar grains used for optical dating at Liang Bua, Indonesia, Quaternary Geochronology, 64, 101182, https://doi.org/10.1016/j.quageo.2021.101182, 2021.

Sontag-González, M. and Fuchs, M.: Spectroscopic investigations of infrared-radiofluorescence (IR-RF) for equivalent dose estimation, Radiation Measurements, 153, 106733, https://doi.org/10.1016/j.radmeas.2022.106733, 2022.

Trautmann, T., Krbetschek, M. R., Dietrich, A., and Stolz, W.: Investigations of feldspar radioluminescence: potential for a new dating technique, Radiation Measurements, 29, 421–425, https://doi.org/10.1016/S1350-4487(98)00012-2, 1998.

**Additional comments from the associate editor (authors' responses are shown in red after each comment):**

When "luminescence" is used but what you actually meant was IR-RF, it is better to replace it with IR-RF (e.g. line 27, 70) to avoid potential confusion.

- We have replaced 5 instances of 'luminescence' with 'IR-RF'. In the remaining ones, we are referring to luminescence in general (i.e., not specifically IR-RF).

Line 43-58: Please make the comparison of dating limit between IRSL and IR-RF using the same (range of) does rate. Also, 600 Gy limit of IRSL seems to be the lowest among the reported values, and I think that both signals have the natural limit of ~1500 Gy, because the lower D0 values of pIRIR were compromised by the test dose.

- Apologies for this inconsistency. We have changed it to (changes in bold type): "allows for the **routine** dating of older deposits of up to **~300 000 years or ~9**00 000 years (considering a dose rate of **3 Gy ka$^{-1}$ or** 1 Gy ka$^{-1}$**, respectively**)"

- We agree that 600 Gy is a very conservative estimate for the pIRIR dating limit. We have revised it to 900 Gy, which is the saturation limit in Liu et al. (2016) (based on 2D$_0$) and is also given as the upper range for the reliable limit in the Murari et al. (2021) literature review (we had previously used the lower range). Test doses of >30% of D$_e$ were used by Liu et al. (2016), but we acknowledge that this is an aspect under current investigation by the luminescence dating community. Since we are aware that several researchers have obtained significantly higher doses under specific circumstances, we have added the word "routine" to the sentence in question.

Liu, J., Murray, A. S., Buylaert, J.-P., Jain, M., Chen, J., and Lu, Y.: Stability of fine-grained TT-OSL and post-IR IRSL signals from a c. 1 Ma sequence of aeolian and lacustrine deposits from the Nihewan Basin (northern China), Boreas, 45, 703–714, https://doi.org/10.1111/bor.12180, 2016.

Line 140: Feldspar mineralogy (mostly anothoclase) and chemistry were reported in detail by Phillips et al (2023) for Gele Tuff. Please add more information sample feldspars here, rather than saying at later stage that the IR-RF characteristic of these samples could be different due to volcanic nature. Also, I think you could have selected K-richer grains by using a lower density heavy liquid, as the end member density of sanidine is 2.52.

- We have added details on the Gele tuff from the suggested paper: "Previous compositional analyses of Gele Tuff pumice feldspars (crushed clasts without density separation) indicate they are mostly composed of anorthoclase with smaller proportions of sanidine and plagioclase; K, Na and Ca contents ranged ~1–6 wt %, ~5–6 wt % and ~0–3 wt %, respectively, without appreciable differences between the grains' cores and rims (Phillips et al., 2023). Relatively high Ba contents of up to 0.8wt% were also reported in that study, with a positive correlation between Ba and Na contents." However, we stress that those results cannot be directly compared to ours because our sample underwent density separation to increase the proportion of K-rich feldspar and their feldspar grains were obtained by crushing pumice clasts and then visually selecting grains.

- In future work, we will consider testing a lower density, thank you for the suggestion.

**Other changes by the authors**

1) Correction of Ar/Ar age of the Gele tuff: "1.315 ± 0.002 Ma (Phillips et al., 2023)."

2) Correction of 'normalised' to 'normalized' in Fig. 8 legend.